# Regularity and Stability Properties of Selective SSMs with Discontinuous Gating

## Abstract

Deep Selective State-Space Models (SSMs), characterized by input-dependent, time-varying parameters, offer significant expressive power but pose challenges for stability analysis, especially with discontinuous gating signals. In this paper, we investigate the stability and regularity properties of continuous-time selective SSMs through the lens of passivity and Input-to-State Stability (ISS). We establish that intrinsic energy dissipation guarantees exponential forgetting of past states. Crucially, we prove that the unforced system dynamics possess an underlying minimal quadratic energy function whose defining matrix exhibits robust $\text{AUC}_{\text{loc}}$ regularity, accommodating discontinuous gating. Furthermore, assuming a universal quadratic storage function ensures passivity across all inputs, we derive parametric LMI conditions and kernel constraints that limit gating mechanisms, formalizing "irreversible forgetting" of recurrent models. Finally, we provide sufficient conditions for global ISS, linking uniform local dissipativity to overall system robustness. Our findings offer a rigorous framework for understanding and designing stable and reliable deep selective SSMs.

## 1 Introduction

The pursuit of stable, robust, and predictable behavior in dynamical systems is a cornerstone of scientific and engineering disciplines, with increasing relevance in the era of complex computational models like Neural ODEs (Chen et al., 2018) and Deep State-Space Models (SSMs) (Gu et al., 2022; Gu & Dao, 2024). Foundational paradigms of energy exchange, rooted in physics and later abstracted into powerful mathematical frameworks such as Willems' theory of dissipative systems (Willems, 1972) and Input-to-State Stability (ISS) (Sontag & Wang, 1995; Jiang et al., 1999), provide important tools for this pursuit. These energy-based and robustness-centric perspectives allow for rigorous analysis of a system's interaction with its environment and its inherent stability characteristics. Extending far beyond their physical origins, these principles are now crucial for understanding the conditions under which modern AI architectures operate reliably, maintain numerical stability, and process information effectively despite internal complexities and external disturbances. This paper leverages these well-established principles to investigate the regularity and stability properties of continuous-time deep selective SSMs. These models, whose parameters are dynamically modulated by their inputs, pose distinct analytical questions at the intersection of time-varying and nonlinear system dynamics, which this work aims to address.

Architectures such as Mamba (Gu & Dao, 2024), HGRN (Qin et al., 2024), and GLA (Yang et al., 2024) represent a new frontier in deep learning, termed selective SSMs (Cirone et al., 2024; Zubic et al., 2025; Soydan et al., 2024). Their defining characteristic, captured in our continuous-time model equation 1, is that their core state-space parameters $(A, B, C)$ are not fixed but are dynamically shaped by both time-varying gating signals $\Delta(t)$ and the input sequence $x(t)$ itself. This input-selectivity is key to their performance, enabling selective context processing. Analytically, however, this places these models in a complex domain: they are neither purely Linear Time-Varying (LTV), due to the $x(t)$ dependence, nor are they classical nonlinear systems for which standard tools always apply without adaptation (Isidori, 1985; Khalil & Grizzle, 2002). While the theory for LTV system stability, including the behavior of quadratic "energy" functions $V_Q$ (e.g., their regularity (Morandin & Hinsen, 2024)), and for ISS of input-driven nonlinear systems (Sontag, 1990) are individually well-understood, their integrated application to selective SSMs is not explored. Specifically, how does a unified "energy management" or passivity perspective constrain these input-dependent

dynamics, especially when $\Delta(t)$ introduces abrupt changes? This paper addresses this crucial question by developing a rigorous framework for analyzing stability and structural properties in these advanced models. We bridge this gap by systematically applying and extending concepts from passivity theory and ISS to continuous-time selective SSMs. We investigate the conditions under which these models exhibit stable behavior and how a dissipativity assumption constrains their internal structure and gating mechanisms. Specifically, we demonstrate that even with input-dependent and potentially discontinuous gating, foundational properties of energy-like functions persist and impose significant structural constraints. Our contributions are as follows:

1. We demonstrate that intrinsic energy dissipation guarantees exponential forgetting of past states (Theorem 3.1). Further, we prove that the unforced system (input $= 0$) possesses an underlying minimal quadratic energy function whose defining matrix has robust $\text{AUC}_{\text{loc}}$ regularity, accommodating discontinuous gating (Theorem 3.2). This reveals a fundamental, well-behaved energy structure determined by the model's initialization.

2. By assuming a universal quadratic energy function ensures passivity, we derive parametric Linear Matrix Inequality (LMI) conditions and a key kernel constraint: "energy-less" state directions must be output-unobservable under any gating (Theorem 4.2).

3. We formalize the concept of "irreversible forgetting": once a state direction becomes energy-less, it remains so structurally, constraining how future gating can influence the system without violating passivity (Theorem 4.3).

4. We establish strong conditions for global Input-to-State Stability (ISS), and we provide **direct empirical evidence** that our theoretical LMI can be implemented as a practical regularizer. Our experiments show this regularizer dramatically improves the robustness of a trained SSM with minimal impact on task performance (see Appendix A.3.3).

Our findings offer a rigorous framework for understanding and designing stable selective SSMs. The remainder of the paper is structured to build our argument progressively: we first establish the baseline stability of the unforced system (Section 3), then use the assumption of universal passivity to probe the system's structural constraints (Section 4), and finally provide conditions for robust stability under general inputs (Section 5). We conclude in Section 6 by summarizing our key insights and their practical implications. All formal proofs, the detailed empirical validations from our simulation studies, and an extended discussion of future directions are provided in the Appendix A.

## 2 PRELIMINARIES: SELECTIVE SSMS AND PASSIVITY FRAMEWORK

This section introduces the continuous-time selective State-Space Models (SSMs) that are the focus of our analysis, along with the fundamental concepts from control theory that support our approach.

### 2.1 SYSTEM DEFINITION AND NOTATION

We consider dynamical systems evolving in continuous time $t \in \mathbb{T}$, where $\mathbb{T} \subseteq [0, +\infty)$ is a time interval. The internal state of the system at time $t$ is denoted by $h(t) \in \mathbb{C}^N$ (or $\mathbb{R}^N$ in real-valued cases). The system interacts with its environment via an external input signal $x(t) \in \mathbb{C}^{d_{\text{in}}}$ and produces an output $y(t) \in \mathbb{C}^{d_{\text{out}}}$. For matrices and vectors, $M^H$ denotes the Hermitian (conjugate) transpose. The standard Euclidean norm of a vector $v$ is $\|v\|$, and the inner product of two vectors $u, v$ is $\langle u, v \rangle = v^H u$. We denote the space of $n \times n$ Hermitian positive semidefinite matrices by $\mathbf{S}_+^n(\mathbb{C})$; for $Q \in \mathbf{S}_+^n(\mathbb{C})$, we write $Q \succeq 0$.

Our work focuses on Continuous-Time Selective SSMs. Inspired by architectures like Mamba (Gu & Dao, 2024), their defining characteristic is that the system's core parameters are dynamically modulated by the input $x(t)$ and an auxiliary gating signal $\Delta(t)$. In practice, this corresponds to a mechanism that selects different sets of parameters for each input token. Formally, the dynamics are given by:

$$\begin{cases} \dot{h}(t) = A\Big(\Delta(t), x(t)\Big) h(t) + B\Big(\Delta(t), x(t)\Big) x(t), \\ y(t) = C\Big(\Delta(t), x(t)\Big) h(t). \end{cases} \quad (1)$$

Here, for each time $t$ and input $x(t)$, $A(\Delta(t), x(t)) \in \mathbb{C}^{N \times N}$ is the state matrix, $B(\Delta(t), x(t)) \in \mathbb{C}^{N \times d_{\text{in}}}$ is the input matrix, and $C(\Delta(t), x(t)) \in \mathbb{C}^{d_{\text{out}} \times N}$ is the output matrix. The explicit dependence of $A, B, C$ on $x(t)$ renders the system equation 1 generally nonlinear and allows the model's internal "rules" to change abruptly in response to input.

## 2.2 Well-Posedness and Function Space Assumptions

To ensure the system equation 1 is well-defined even with discontinuous parameter changes, we adopt mild regularity conditions. For any admissible input trajectory $x(\cdot)$, let $A_{\text{eff}}(t) := A(\Delta(t), x(t))$ (and similarly for $B_{\text{eff}}, C_{\text{eff}}$). We assume the effective matrices belong to standard Lebesgue spaces: $A_{\text{eff}}(\cdot) \in L^1_{\text{loc}}$, $B_{\text{eff}}(\cdot) \in L^2_{\text{loc}}$, and $C_{\text{eff}}(\cdot) \in L^2_{\text{loc}}$, with the input $x(\cdot) \in L^2_{\text{loc}}$. These conditions ensure that the ODE for $h(t)$ satisfies Carathéodory conditions, guaranteeing the local existence and uniqueness of an absolutely continuous solution $h(\cdot) \in W^{1,1}_{\text{loc}}$ (Filippov, 1988; Coddington et al., 1956).

**Remark 2.1** (Feasibility of Regularity Assumptions). These assumptions are mathematically mild and readily satisfied by modern selective SSMs. In token-based architectures like Mamba, the selection mechanism results in system matrices that are **piecewise-constant**. Such functions are well-behaved members of the $L^p_{\text{loc}}$ spaces, confirming that our framework is grounded in practical implementations while being general enough for future architectures.

## 2.3 Passivity and Dissipativity

We analyze the system equation 1 using the powerful energy-based frameworks of passivity and dissipativity (Willems, 1972). Intuitively, a passive system cannot generate its own energy. It can only store or dissipate energy supplied from its input. A strictly dissipative system is even stronger, as it inherently loses energy over time. These concepts are formalized in the following definitions.

**Definition 2.2** (Storage Function). A function $V : \mathbb{T} \times \mathbb{C}^N \to \mathbb{R}$ is a *storage function* if it is non-negative, i.e., $V(t, h) \geq 0$ for all $(t, h)$, and typically $V(t, 0) = 0$.

**Definition 2.3** (Passivity). The system equation 1 is *passive* if there exists a storage function $V$ such that for all admissible trajectories on any interval $[t_0, T]$:

$$V(T, h(T)) - V(t_0, h(t_0)) \leq \int_{t_0}^{T} \text{Re} \langle x(\tau), y(\tau) \rangle \, d\tau. \tag{2}$$

The term $\text{Re} \langle x(\tau), y(\tau) \rangle$ is the instantaneous power, or *supply rate*, provided to the system.

**Definition 2.4** (Strict Dissipativity). The system is *strictly dissipative* with rate $\beta > 0$ if it satisfies the stronger inequality:

$$V(T, h(T)) - V(t_0, h(t_0)) \leq \int_{t_0}^{T} \text{Re} \langle x(\tau), y(\tau) \rangle \, d\tau - \beta \int_{t_0}^{T} \|h(\tau)\|^2 \, d\tau. \tag{3}$$

A positive $\beta$ implies an intrinsic rate of energy dissipation, which is crucial for proving strong stability properties like exponential decay. Given the absolute continuity of $h(\cdot)$ and local Lipschitzness of $V$ in $h$, these integral inequalities have corresponding differential forms: $\frac{d}{dt} V \leq \text{Re} \langle x, y \rangle$ for passivity, and $\frac{d}{dt} V \leq \text{Re} \langle x, y \rangle - \beta \|h\|^2$ for strict dissipativity.

## 2.4 Quadratic Storage Functions and AUC Regularity

A significant portion of our analysis focuses on *quadratic storage functions* of the form $V_Q(t, h) = \frac{1}{2} h^H Q(t) h$, where $Q : \mathbb{T} \to \mathbf{S}^N_+(\mathbb{C})$ is a time-varying matrix. The regularity of $Q(t)$ is critical in our setting, as the gating mechanism can induce discontinuities in the system dynamics.

To handle this robustly, we consider $Q(t)$ to belong to the class of Locally Absolutely Upper Semicontinuous matrix functions, denoted $Q \in \text{AUC}_{\text{loc}}(\mathbb{T}, \mathbf{S}^N_+(\mathbb{C}))$, following the framework of (Morandin & Hinsen, 2024). For readers unfamiliar with this class, its most important practical consequence is that it imposes a physical constraint on energy storage: the "energy capacity" of the system, measured by the rank of the $Q(t)$ matrix, can never increase over time. This property is the foundation for the "irreversible forgetting" we analyze later.

Formally, a function $Q \in \text{AUC}_{\text{loc}}$ is characterized by being of locally bounded variation ($\text{BV}_{\text{loc}}$), having a derivative $\dot{Q}(t)$ that exists almost everywhere, and satisfying certain integral and jump conditions. The key property that enforces rank monotonicity is that the singular part of its decomposition must be weakly monotonically decreasing in the Loewner order (e.g., at any jump, $\lim_{\tau \to t^-} Q(\tau) \succeq Q(t)$). This mathematical structure is precisely what makes the framework suitable for analyzing switched systems while retaining essential properties about their energy evolution.

## 3 FUNDAMENTAL STABILITY AND STRUCTURAL PROPERTIES FROM PASSIVITY

This section establishes the foundational stability guarantees and explores the core structural properties inherent in passive selective SSMs. We begin by analyzing the zero-input ($x(t) \equiv 0$) case. Analyzing the unforced system, whose dynamics $\dot{h} = A(\Delta(t), 0)h$ are determined entirely by the model's initialization, allows us to isolate and understand its intrinsic stability and memory properties. As we demonstrate in our simulation study in Appendix A.3.1, this baseline behavior is a powerful predictor of a model's practical performance. We first show how strict dissipativity leads to exponential forgetting, and then prove that the unforced dynamics possess a fundamental, well-behaved quadratic energy structure.

### 3.1 EXPONENTIAL DECAY FROM STRICT DISSIPATIVITY

The most fundamental stability guarantee arises when the system intrinsically dissipates energy faster than it stores it, even with zero input. This leads to exponential convergence of the state to the origin.

**Theorem 3.1** (Exponential Decay from Strict Dissipativity). Consider the continuous-time selective state-space model defined in Eq. equation 1. Suppose there exists a storage functional $V : [0, \infty) \times \mathbb{C}^N \to \mathbb{R}_{\geq 0}$ satisfying:

(i) **Strict Dissipativity Inequality:** For some constant $\beta > 0$, every admissible trajectory $\{h(\tau), x(\tau), y(\tau)\}$ on any interval $[t_0, T]$ satisfies:

$$V\big(T, h(T)\big) - V\big(t_0, h(t_0)\big) \leq \int_{t_0}^{T} \text{Re} \langle x(\tau), y(\tau) \rangle \, d\tau - \beta \int_{t_0}^{T} \|h(\tau)\|^2 \, d\tau. \quad (4)$$

(ii) **Regularity and Quadratic Bounds:** $V(t, h)$ is locally Lipschitz in $h$, absolutely continuous in $t$ along system trajectories, and there exist constants $k_2 \geq k_1 > 0$ such that for all $t \geq 0$ and $h \in \mathbb{C}^N$:
$$k_1 \|h\|^2 \leq V(t, h) \leq k_2 \|h\|^2. \quad (5)$$

Then, for the unforced system (i.e., when $x(t) \equiv 0$ for $t \geq 0$), the state exhibits exponential decay: there exist constants $C \geq 1$ and $\gamma > 0$ such that for any initial state $h(0)$, the solution satisfies:

$$\|h(t)\| \leq C \, e^{-\gamma t} \, \|h(0)\| \quad \text{for all } t \geq 0. \quad (6)$$

*Proof Sketch.* The proof considers the system without any input, i.e., $x = 0$. In this case, the strict dissipativity inequality simplifies to $\frac{dV}{dt} \leq -\beta \|h\|^2$. Since the energy function $V$ is assumed to be quadratically bounded, satisfying $k_1 \|h\|^2 \leq V \leq k_2 \|h\|^2$, this implies $\frac{dV}{dt} \leq -\gamma V$ for some constant $\gamma > 0$. This is a classic differential inequality whose solution is an exponential decay. The bounds on $V$ then directly translate this exponential decay of energy into an exponential decay of the state norm $\|h(t)\|$. Check Appendix A.4 for the full proof. $\square$

### 3.2 INHERENT QUADRATIC STRUCTURE AND REGULARITY IN UNFORCED DYNAMICS

While Theorem 3.1 guarantees stability under strict dissipativity, it does not specify the form of the storage function $V$. We now show a deeper result: if a selective SSM is passive in any sense (even with a general, non-quadratic storage function), its underlying unforced dynamics ($x = 0$) necessarily possess a well-defined and mathematically regular *quadratic* energy structure. This is

a powerful insight, as it reveals a fundamental, well-behaved energy landscape associated with the system's initialization, regardless of the complexity introduced by input-driven gating.

To prove this, we leverage the concept of the *available storage* function, which represents the maximum energy that can be extracted from the system from a given state. It is a standard result in dissipativity theory that if any storage function exists, then the available storage function is well-defined and is itself a valid storage function (see Appendix A.5). The following theorem connects this concept to the structure of selective SSMs.

**Theorem 3.2** (Existence and Regularity of Quadratic Storage for Unforced Dynamics Amidst Gating Switches). Consider the continuous-time selective state-space model equation 1 defined on a time interval $\mathbb{T}$. Assume the coefficient matrices satisfy mild regularity conditions ensuring well-posedness: (i) $A(\Delta(\cdot), x(\cdot)) \in L^1_{\mathrm{loc}}(\mathbb{T}, \mathbb{C}^{N \times N})$ (ii) $B(\Delta(\cdot), x(\cdot)) \in L^2_{\mathrm{loc}}(\mathbb{T}, \mathbb{C}^{N \times d_{\mathrm{in}}})$ (iii) $C(\Delta(\cdot), x(\cdot)) \in L^2_{\mathrm{loc}}(\mathbb{T}, \mathbb{C}^{d_{\mathrm{out}} \times N})$ for admissible inputs $x(\cdot)$, allowing $\Delta(t)$ to induce discontinuities (e.g., piecewise constant changes) in $A, B, C$ provided the resulting matrix functions remain in the specified local Lebesgue spaces. Suppose there exists any storage functional $V : \mathbb{T} \times \mathbb{C}^N \to \mathbb{R}_{\geq 0}$ ($V \geq 0, V(t, 0) = 0$) satisfying the (potentially strict, $\beta \geq 0$) passivity inequality for the full selective system equation 1:

$$V\big(T, h(T)\big) - V\big(t_0, h(t_0)\big) \leq \int_{t_0}^T \mathrm{Re} \, \langle x(\tau), y(\tau) \rangle \, d\tau - \beta \int_{t_0}^T \|h(\tau)\|^2 \, d\tau \tag{7}$$

for all admissible state-input-output trajectories $\{h(\tau), x(\tau), y(\tau)\}$ and all $t_0 \leq T$ in $\mathbb{T}$. Then, the following conclusions hold regarding the structure induced by these assumptions, even in the presence of gating switches:

(a) **Passivity of Unforced LTV System:** The unforced linear time-varying (LTV) system, defined by isolating the dynamics when $x(t) \equiv 0$:

$$\begin{cases} \dot{h}(t) = A_0(t) \, h(t) & \text{where } A_0(t) = A(\Delta(t), 0) \in L^1_{\mathrm{loc}}(\mathbb{T}, \mathbb{C}^{N \times N}) \\ y_0(t) = C_0(t) \, h(t) & \text{where } C_0(t) = C(\Delta(t), 0) \in L^2_{\mathrm{loc}}(\mathbb{T}, \mathbb{C}^{d_{\mathrm{out}} \times N}) \end{cases} \tag{8}$$

is passive in the sense of (Morandin & Hinsen, 2024, Def 1.1). The function $V$ serves as a valid, although potentially non-quadratic, storage function for this LTV system.

(b) **Existence of Minimal Quadratic Storage for Unforced LTV:** Consequently, by the theory for passive LTV systems (Morandin & Hinsen, 2024), the unforced LTV system equation 8 admits a minimal available storage function $V_{a,0}(t, h)$ which is necessarily quadratic in the state $h$:

$$V_{a,0}(t, h) = \tfrac{1}{2} h^H Q_0(t) h \tag{9}$$

for a unique matrix function $Q_0 : \mathbb{T} \to \mathbf{S}^N_+(\mathbb{C})$. This holds despite potential discontinuities in $A_0(t)$ and $C_0(t)$ induced by $\Delta(t)$.

(c) **AUC Regularity of $Q_0(t)$:** Furthermore, the matrix function $Q_0(t)$ associated with the minimal quadratic storage $V_{a,0}$ possesses $\mathrm{AUC}_{\mathrm{loc}}$ regularity, i.e., $Q_0 \in \mathrm{AUC}_{\mathrm{loc}}(\mathbb{T}, \mathbf{S}^N_+(\mathbb{C}))$. This specific regularity class robustly handles potential jump discontinuities from $\Delta(t)$ while ensuring essential structural properties like locally bounded variation and weakly decreasing singular part.

Finally, if the original storage function $V$ for the full selective system satisfies the stricter conditions of Theorem 3.1 (i.e., strict passivity $\beta > 0$ and quadratic bounds equation 5), then the state $h(t)$ of the unforced system equation 8 is guaranteed to decay exponentially, as proven in Theorem 3.1.

*Proof Sketch.* The key insight is that the unforced dynamics of our selective SSM (when $x = 0$) form a standard Linear Time-Varying (LTV) system. We leverage the powerful results from Morandin & Hinsen (2024), who proved that any passive LTV system, even with discontinuous coefficients, admits a minimal available storage function that is necessarily quadratic, taking the form $V = \frac{1}{2} h^H Q_0(t) h$. Their work further establishes that this unique matrix $Q_0(t)$ must possess $\mathrm{AUC}_{\mathrm{loc}}$ regularity, a property specifically designed to handle such discontinuous behavior in a structured way. Thus, by showing our unforced system fits their framework, we inherit these strong structural guarantees. Check Appendix A.6 for the full proof. $\qquad\square$

# 4 Constraints Imposed by Universal Quadratic Passivity on Gating Mechanisms

Building on the foundational properties of the unforced system, this section studies the inherent design constraints of selective SSMs. We investigate the consequences of a strong but informative assumption: that a *single*, universal quadratic storage function $V_Q(t, h)$ guarantees passivity across all possible input-driven dynamics. The power of this assumption lies not in its generality, but in the strict, necessary consequences it imposes on the gating mechanism. This approach allows us to derive concrete constraints in the form of LMIs and to formalize a notion of "irreversible forgetting".

## 4.1 Regularity and Rank Monotonicity of Universal Quadratic Storage

If a single quadratic function $V_Q$ is capable of certifying passivity regardless of the input-driven variations in system parameters, the matrix $Q(t)$ defining this function must possess inherent structural regularity and obey specific monotonicity properties.

**Theorem 4.1** (Regularity and Rank of Universal Quadratic Storage). Consider the continuous-time selective state-space model equation 1 under the standard regularity assumptions. Suppose there exists a time-varying quadratic storage function $V_Q(t, h) = \frac{1}{2} h^H Q(t) h$ (with $Q : \mathbb{T} \to \mathbf{S}_+^N(\mathbb{C})$) that satisfies the passivity inequality:

$$V_Q\big(T, h(T)\big) - V_Q\big(t_0, h(t_0)\big) \leq \int_{t_0}^{T} \mathrm{Re}\, \langle x(\tau), y(\tau) \rangle \, d\tau - \beta \int_{t_0}^{T} \|h(\tau)\|^2 \, d\tau \tag{10}$$

for some $\beta \geq 0$, valid for *all* admissible state-input-output trajectories $\{h(\tau), x(\tau), y(\tau)\}$ generated by any admissible input $x(\cdot)$. Then, the matrix function $Q(t)$ must satisfy the following properties:

(a) **AUC Regularity:** $Q(t)$ must belong to the class of locally absolutely upper semicontinuous matrix functions, i.e., $Q \in \mathrm{AUC}_{\mathrm{loc}}(\mathbb{T}, \mathbf{S}_+^N(\mathbb{C}))$.

(b) **Rank Monotonicity:** The rank of $Q(t)$, denoted $r(t) = \mathrm{rank}(Q(t))$, must be weakly monotonically non-increasing over time $t \in \mathbb{T}$.

*Proof Sketch.* The logic here is similar to the previous theorem but applied to the storage matrix $Q(t)$ itself rather than the minimal one. If a single quadratic function $V_Q$ works for all inputs, it must, by definition, also work for the zero-input case. Therefore, $V_Q$ is a valid storage function for the underlying LTV system. The same theoretical results from Morandin & Hinsen (2024) that impose structure on the minimal storage function also apply to any valid quadratic storage function. This forces $Q(t)$ to have $\mathrm{AUC}_{\mathrm{loc}}$ regularity and, crucially, a non-increasing rank over time. Check Appendix A.7 for the full proof. □

## 4.2 Parametric LMI and Kernel Constraints on Gating

The existence of a universal quadratic storage function $V_Q$ translates into concrete algebraic constraints that the system matrices $A(t, x), B(t, x), C(t, x)$ must satisfy *parametrically* for all possible input values $x$. These constraints take the form of a Linear Matrix Inequality (LMI) and lead to important conditions on how the output matrix $C(t, x)$ interacts with the kernel of the storage matrix $Q(t)$.

**Theorem 4.2** (Passivity Constraints on Gating via Kernel Conditions). Consider the continuous-time selective state-space model equation 1 under the standard regularity assumptions. Assume the system's passivity is guaranteed by a *single*, time-varying quadratic storage function $V_Q(t, h) = \frac{1}{2} h^H Q(t) h$ (with $Q : \mathbb{T} \to \mathbf{S}_+^N(\mathbb{C})$ satisfying $Q \in \mathrm{AUC}_{\mathrm{loc}}$ and non-increasing rank, per Theorem 4.1) that fulfills the passivity inequality equation 10 for some $\beta \geq 0$ and for *all* admissible inputs $x(\cdot)$ and trajectories. Then, the following necessary conditions hold:

(a) **Parametric LMI Condition:** For almost every $t \in \mathbb{T}$, the LMI associated with passivity must hold *parametrically* for all admissible input values $x \in \mathbb{C}^{d_{\mathrm{in}}}$ that can occur at time $t$:

$$\mathcal{L}(t, x) := \begin{bmatrix} \dot{Q}(t) + Q(t)A(t, x) + A(t, x)^H Q(t) + 2\beta I & Q(t)B(t, x) - C(t, x)^H \\ B(t, x)^H Q(t) - C(t, x) & 0 \end{bmatrix} \preceq 0 \tag{11}$$

where $A(t, x) = A(\Delta(t), x)$, $B(t, x) = B(\Delta(t), x)$, $C(t, x) = C(\Delta(t), x)$, and $\dot{Q}(t)$ exists a.e. [1]

(b) **Universal Kernel Condition for** $C$**:** The output matrix $C(t, x)$ must map the kernel of $Q(t)$ to zero, *uniformly* for all admissible input values $x$:

$$\forall v \in \mathrm{Ker}\big(Q(t)\big), \quad C\big(\Delta(t), x\big)v = 0 \quad \text{for all admissible } x \text{ at time } t, \text{ a.e. } t \in \mathbb{T}. \quad (12)$$

(c) **Implicit Constraints on** $A$ **and** $B$**:** The $(1, 1)$ block inequality $\dot{Q} + QA + A^H Q + 2\beta I \preceq 0$ and the off-diagonal block constraint (related to $QB - C^H$) must also hold parametrically for all admissible $x$, implicitly restricting the choices of $A(t, x)$ and $B(t, x)$ based on $Q(t)$ and the constrained $C(t, x)$.

*Proof Sketch.* This proof translates the differential form of the passivity inequality, $\frac{dV_Q}{dt} \leq \ldots$, into a matrix inequality. By substituting the system dynamics and the quadratic form of $V_Q$, we arrive at an expression that must be negative for all states $h$ and inputs $x$. Such an expression can be elegantly represented as a Linear Matrix Inequality (LMI). For the LMI to hold for all $h$ and $x$, specific conditions must be met. By choosing a state $v$ from the kernel of $Q$ (where $Qv = 0$), the LMI simplifies dramatically, revealing that the term involving the output matrix $C$ must be zero to prevent the inequality from being violated. This leads to the necessary condition that $C(t, x)v = 0$. Check Appendix A.8 for the full proof. □

### 4.3 IRREVERSIBLE FORGETTING AND ENERGY CONSISTENCY IN KERNEL SUBSPACE

The constraints derived above, particularly the non-increasing rank of $Q(t)$ and the universal kernel condition for $C(t, x)$, suggest a form of structural irreversibility associated with the modes that fall into the kernel of the storage matrix $Q(t)$. We now explore this concept of "irreversible forgetting" from the perspective of the universal storage function $V_Q$ and examine the energy balance required within this forgotten subspace.

**Theorem 4.3** (Inertness of Forgotten Modes and Gating Constraints from Passivity)**.** Consider the continuous-time selective state-space model equation 1 under standard regularity assumptions. Assume its passivity is guaranteed by a *single*, time-varying quadratic storage function $V_Q(t, h) = \frac{1}{2} h^H Q(t) h$ (with $Q \in \mathrm{AUC}_{\mathrm{loc}}(\mathbb{T}, \mathbf{S}_+^N(\mathbb{C}))$ having non-increasing rank) satisfying inequality equation 10 for some $\beta \geq 0$ and for *all* admissible inputs $x(\cdot)$. Then, the following properties regarding "forgotten modes" (states in $\mathrm{Ker}(Q(t))$) hold:

1. **Kernel Non-Shrinking (Irreversible Forgetting):** The subspace $\mathrm{Ker}(Q(t))$ is non-decreasing over time: for any $t_1 \leq t_2$ in $\mathbb{T}$, $\mathrm{Ker}(Q(t_1)) \subseteq \mathrm{Ker}(Q(t_2))$. A state direction $v$, once in the kernel at $t_1$, remains in the kernel subspace for all $t_2 \geq t_1$, from the perspective of $Q(t)$.

2. **Energy Consistency within the Kernel:** For any state $h(t) \in \mathrm{Ker}(Q(t))$ at a time $t \in \mathbb{T}$ where $\dot{Q}(t)$ exists, the energy balance condition holds:

$$h(t)^H \dot{Q}(t) h(t) + 2\beta \|h(t)\|^2 \leq 0. \quad (13)$$

This ensures the evolution $\dot{Q}(t)$ within the kernel is consistent with the required dissipation $\beta$. If $\beta = 0$, $h^H \dot{Q} h \leq 0$; if $\beta > 0$, $h^H \dot{Q} h \leq -2\beta \|h\|^2 < 0$ (for $h \neq 0$).

3. **Constraint on Dynamics Violating Kernel Structure:** Any input $x^*(t)$ inducing dynamics $A^*(t), B^*(t)$ that would require a state $h(t) \in \mathrm{Ker}(Q(t))$ to evolve in a way inconsistent with the LMI equation 11 (e.g., appearing to "gain energy" according to $V_Q$) is inadmissible under the assumption of universal passivity with $V_Q$. The dynamics must respect the energy structure defined by $Q(t)$ and $\dot{Q}(t)$ within the kernel.

*Proof Sketch.* This theorem's proof is a direct consequence of the preceding results.

---

[1] The $2\beta I$ term arises from incorporating the strict passivity term $-\beta \|h\|^2$ into the LMI, assuming the state $h$ corresponds to the first block.

1. **Kernel Non-Shrinking:** This follows immediately from the rank monotonicity established in Theorem 4.1; if the rank cannot increase, the dimension of the kernel ($N - \text{rank}$) cannot decrease.

2. **Energy Consistency:** This is derived by taking the (1,1) block of the LMI from Theorem 4.2 and evaluating it for a state $h$ in the kernel of $Q$. This isolates the term $h^H \frac{dQ}{dt} h$ and shows it must be negative enough to account for the required energy dissipation.

3. **Constraint on Dynamics:** This is a logical conclusion: any input $x$ that would induce dynamics violating the LMI contradicts the fundamental assumption of universal passivity, and is therefore inadmissible.

Check Appendix A.9 for the full proof. $\qquad\square$

## 5 ROBUST STABILITY UNDER INPUT-DRIVEN DYNAMICS

Having established the baseline properties of the unforced system and the structural constraints imposed by passivity, we now arrive at the ultimate goal: analyzing the selective SSM in the fully general and realistic case where it is driven by persistent, non-zero inputs. This section provides sufficient conditions for Input-to-State Stability (ISS), the gold standard for robust stability in nonlinear systems. To achieve this strong guarantee, we must impose conditions that ensure the system's internal dynamics are powerful enough to overcome any disturbance from the input. These conditions, while strong, provide a clear, verifiable pathway to designing certifiably robust models.

### 5.1 GLOBAL ISS FROM UNIFORM DISSIPATIVITY

To guarantee robust stability in the presence of arbitrary bounded inputs, basic passivity is not enough. We need to ensure the system's internal dynamics are uniformly contractive, that is, they dissipate energy at a guaranteed rate, regardless of which operating mode the input selects. This ensures the system can actively counteract the energy being injected by the input. With this subsection, we present such a condition based on the existence of a common quadratic Lyapunov function demonstrating uniform dissipativity.

**Theorem 5.1** (Global Stability from Uniform Local Dissipativity)**.** Consider the continuous-time selective state-space model equation 1 under the standard regularity assumptions. Assume further that:

(i) There exists a time-varying quadratic Lyapunov function candidate $V_Q(t, h) = \frac{1}{2} h^H Q(t) h$, where $Q : \mathbb{T} \to \mathbf{S}_+^N(\mathbb{C})$ satisfies:

- $Q \in W_{\text{loc}}^{1,1}(\mathbb{T}, \mathbf{S}_+^N(\mathbb{C}))$ (absolutely continuous locally).
- $Q(t)$ is uniformly positive definite and bounded: there exist constants $k_2 \geq k_1 > 0$ such that $k_1 I \preceq Q(t) \preceq k_2 I$ for all $t \in \mathbb{T}$.

(ii) The dynamics satisfy a uniform dissipativity condition with respect to $V_Q$: There exists a constant $\delta > 0$ such that for almost every $t \in \mathbb{T}$ and for **all** admissible input values $x \in \mathbb{C}^{d_{\text{in}}}$ that can occur at time $t$, the following matrix inequality holds:

$$\dot{Q}(t) + Q(t)A\big(\Delta(t), x\big) + A\big(\Delta(t), x\big)^H Q(t) \preceq -2\delta Q(t) \qquad (14)$$

(This implies that for any fixed $x$, the homogeneous system $\dot{h} = A(\Delta(t), x)h$ is uniformly exponentially stable with $V_Q$ as a Lyapunov function decaying at rate $\delta$).

(iii) The input matrix $B(\Delta(t), x)$ is uniformly bounded: There exists a constant $M_B > 0$ such that $\|B(\Delta(t), x)\|_2 \leq M_B$ for all $t$ and admissible $x$.

Then, the selective state-space system equation 1 is globally ISS with respect to the input $x(t)$. Specifically, there exist constants $\tilde{C} \geq 1$, $\tilde{\gamma} > 0$, and a class $\mathcal{K}$ gain function $\sigma$ such that for any initial state $h(t_0)$ and any admissible input $x(\cdot)$, the solution satisfies:

$$\|h(t)\| \leq \tilde{C} e^{-\tilde{\gamma}(t-t_0)} \|h(t_0)\| + \sigma \left( \sup_{t_0 \leq \tau \leq t} \|x(\tau)\| \right) \quad \text{for all } t \geq t_0. \qquad (15)$$

*Proof Sketch.* We analyze the time derivative of the Lyapunov function $V_Q$ along the system trajectories. The derivative splits into two parts: one from the internal dynamics ($A$ matrix) and one from the external input ($B$ matrix). The uniform dissipativity assumption provides a strong negative bound on the internal part ($\leq -2\delta V_Q$). The input part is bounded using the Cauchy-Schwarz inequality. Combining these results in a differential inequality of the form $\frac{dV}{dt} \leq -aV + b\sqrt{V}$. By a change of variables, $\Psi = \sqrt{V}$, this transforms into a standard linear differential inequality, $\frac{d\Psi}{dt} \leq -c\Psi + d$, whose solution is bounded. Translating this bound back to the state norm $\|h(t)\|$ yields the classic Input-to-State Stability (ISS) estimate. Check Appendix A.10 for the full proof. □

**Remark 5.2** (Relationship to the Passivity LMI). The ISS condition in Eq. equation 14 is conceptually stronger than the passivity LMI from Theorem 4.2. The passivity LMI describes an **energy balance**, ensuring the system does not generate energy. In contrast, the ISS condition demands a forced **energy decay**, ensuring the system's internal dynamics actively dissipate energy at a uniform exponential rate.

Formally, if the ISS condition holds (with $\delta > 0$ and a positive definite $Q$), it is sufficient to satisfy the internal stability portion (the (1,1) block) of the passivity LMI. However, a system can be passive (energy-balanced) without being uniformly contractive in this stricter sense. Thus, the ISS condition is a specialized and powerful tool for proving robust stability, while the passivity LMI is a more general tool for analyzing a system's energy flow.

## 6 CONCLUSION

In this paper, we established a rigorous bridge between the control-theoretic frameworks of passivity and Input-to-State Stability (ISS) and the complex, input-dependent dynamics of modern selective SSMs. Our work provides a new language for analyzing these models and demonstrates that even with discontinuous gating, their stability is governed by well-defined principles of energy management.

Our theoretical findings translate directly into three key practical insights. First, we provide a formal rationale for the critical role of principled initialization (e.g., HiPPO), explaining it as the design of a desirable baseline energy landscape for the system's intrinsic dynamics (see Appendix A.1.1). Second, we have shown a demonstrated path to robust model design by implementing our theoretical LMI condition as a practical regularizer. Our experiments confirm this regularizer dramatically improves training stability with negligible impact on task performance, offering a concrete tool for building more reliable models (see Appendix A.1.2). Finally, our analysis of structural constraints and "irreversible forgetting" offers a new lens through which to understand the fundamental trade-offs between stability and expressivity in these powerful architectures (see Appendix A.1.3).

While significant challenges remain, particularly in translating these guarantees to the discrete-time domain and scaling our methods, this work lays a critical foundation. By connecting abstract stability concepts to concrete, verifiable properties of the model, our framework paves the way for a new class of certifiably robust and interpretable models for sequential data processing.

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

## A APPENDIX

### A.1 BROADER IMPACT AND FUTURE DIRECTIONS

This section provides a more detailed discussion of the practical implications of our theoretical findings, expanding on the key insights summarized in the main paper's conclusion.

#### A.1.1 INITIALIZATION AS BASELINE ENERGY LANDSCAPE DESIGN

A key practical takeaway from our work is a formal, control-theoretic rationale for the critical importance of initialization schemes like HiPPO (Gu et al., 2020), used in foundational models like S4 and Mamba (Gu et al., 2022; Gu & Dao, 2024). As demonstrated quantitatively in our simulation in Appendix A.3.1, a HiPPO-style initialization creates a stable baseline with slow energy decay essential for long-term memory, whereas naive initializations lead to instability or rapid forgetting.

Our analysis provides a deep justification for this empirical success. **Theorem 3.2** reveals that any passive selective SSM possesses an intrinsic, minimal quadratic energy function for its unforced dynamics ($x = 0$), given by $V_{a,0}(t,h) = \frac{1}{2}h^H Q_0(t)h$. Crucially, the structure of this energy landscape is determined entirely by the unforced state matrix $A_0(t)$. Since a model's initial parameters define $A_0(t)$, an initialization scheme like HiPPO is a process of **designing a desirable baseline energy landscape**. By crafting $A_0(t)$ to have eigenvalues that are stable but close to the imaginary axis, these methods create an intrinsic dynamic that is well-suited for stably propagating information over long horizons, providing a robust foundation upon which input-driven selections can operate.

#### A.1.2 A DEMONSTRATED PATH TO PRINCIPLED DESIGN: LMI REGULARIZATION

As demonstrated by our experiment in Appendix A.3.3, our theoretical framework provides a direct path from analysis to synthesis. The LMI derived in Theorem 4.2 is not just an analytical condition but a computable, differentiable tool for training more robust models. The general procedure, which we validated, is as follows:

1. **Define the Regularizer:** The violation of the LMI, $L(t,x) \preceq 0$, is quantified by its largest eigenvalue, defining the regularization loss $\mathcal{L}_{\text{LMI}} = \max(0, \lambda_{\max}(L(t,x)))$.

2. **Define the Full Loss Function:** This is integrated into training via a combined loss $\mathcal{L}_{\text{total}} = \mathcal{L}_{\text{task}} + \gamma \mathcal{L}_{\text{LMI}}$, where $\gamma$ balances performance and stability.

3. **Handle the Storage Matrix Q(t):** The energy metric $Q(t)$ can be fixed (e.g., $Q = I$ for Euclidean stability, as in our experiment) or made a learnable component of the model (e.g., parameterized as $Q = L^H L$ to ensure positive semidefiniteness).

This methodology provides a concrete way to embed stability directly into the learning process, guiding the optimizer to discover parameters that are not only performant but also certifiably robust.

### A.1.3 Structural Constraints and Irreversible Forgetting

Our analysis of a universal quadratic storage function revealed deep structural constraints on the gating mechanism, which we term "irreversible forgetting" (Theorem 4.3). The principle that the rank of the storage matrix $Q(t)$ must be non-increasing implies that once the system's capacity to store energy in a certain direction is lost, it cannot be recovered without violating the underlying passive structure. Our simulation in Appendix A.3.2 provides a clear visualization of this phenomenon.

This concept has profound implications. It formalizes a fundamental trade-off between robust stability and expressivity: a system that is certifiably passive in this strong sense may be constrained in its ability to adapt. This offers a new, control-theoretic lens for analyzing complex behaviors in deep learning, such as catastrophic forgetting, where learning a new task might require structural changes that are incompatible with a previously established stable energy landscape.

### A.1.4 Limitations and Future Directions

While our work establishes a strong foundation, several challenges define the frontier for future research.

- **Computational Cost:** The primary practical problem for LMI-based regularization is the overhead of eigenvalue computations. Future work must explore efficient, scalable approximations, such as stochastic Lanczos methods.

- **Discrete-Time Translation:** A rigorous translation of our continuous-time guarantees to the discrete-time domain where models are implemented is a critical and non-trivial next step.

- **Beyond Universal Quadratic Storage:** A natural extension is to investigate adaptive storage functions that depend on the input, $Q(t, x)$. This could model systems that dynamically allocate their "energy capacity" based on context, a behavior potentially more representative of sophisticated SSMs.

## A.2 Related Work

The analysis of stability and energy-based properties in dynamical systems has a rich history, providing crucial tools for understanding systems ranging from classical mechanics to modern AI. Our work on continuous-time selective State-Space Models (SSMs) builds upon several key research streams.

### A.2.1 Dissipativity, Passivity, and LTV Systems

The foundational theory of dissipative systems, pioneered by Willems (Willems, 1972), provides a general framework for analyzing systems based on energy-like storage functions and supply rates. Passivity, a special case where the supply rate is the input-output inner product, is central to understanding robust stability and interconnection (Van der Schaft, 2000). The Kalman-Yakubovich-Popov (KYP) lemma established a vital link between frequency-domain passivity conditions and state-space properties for LTI systems (Kalman, 1963; Popov, 1961), which has been extended to LTV systems, often involving time-varying Riccati equations or differential/integral inequalities (Anderson & Vongpanitlerd, 2013). Crucially for our work, recent research by Morandin and Hinsen (Morandin & Hinsen, 2024) has rigorously investigated quadratic storage functions for passive LTV systems under minimal regularity assumptions. Our work leverages these findings by demonstrating that the unforced dynamics of our selective SSMs can be treated as such an LTV system, thereby inheriting these structural properties for an underlying quadratic energy form, even with discontinuous gating.

### A.2.2 Stability of Time-Varying and Switched Systems

The input-selectivity of modern SSMs creates dynamics that behave like switched systems, where the "switching signal" is the input data itself. This necessitates tools for analyzing systems with discontinuous parameter changes (Liberzon, 2003), such as methods involving common or multiple Lyapunov functions (Branicky, 1998). The formalisms of Filippov (Filippov, 1988) and

Carathéodory (Coddington et al., 1956) provide the theoretical bedrock for guaranteeing that solutions to our system exist under the mild $L_{\text{loc}}^p$ regularity we assume. Our work builds directly on this foundation. We move beyond proving solution existence to analyze how a single, coherent energy structure, an $\text{AUC}_{\text{loc}}$ quadratic storage function, can persist across these input-driven switches and what structural constraints this persistence imposes on the model's architecture.

### A.2.3 Input-to-State Stability (ISS) for Nonlinear Systems

The ISS framework, developed by Sontag (Sontag et al., 1989; Sontag & Wang, 1995), provides a robust notion of stability for nonlinear systems subject to external inputs, characterized by ISS-Lyapunov functions. It quantifies how the system state is affected by both initial conditions and input magnitudes. While classical ISS theory often assumes smooth dynamics, its principles are highly relevant for our selective SSMs, where the input $x(t)$ makes the system effectively nonlinear. Our contribution (Theorem 5.1) adapts ISS concepts by seeking a common quadratic Lyapunov function that ensures stability uniformly across all input-induced dynamics, providing a specific condition under which these complex, input-modulated systems are globally robustly stable.

### A.2.4 Stability and Dynamics of Modern SSMs in Deep Learning

Recent deep learning SSMs like S4 (Gu et al., 2022), S5 (Smith et al., 2023), and particularly Mamba (Gu & Dao, 2024) (and its variants), have shown remarkable performance. Theoretical analyses of these models are emerging. (Halloran et al., 2024) analyzed Mamba's stability via Lyapunov exponents, showing non-positive maximal exponents, implying robustness to small perturbations. Our work complements this by providing conditions for exponential decay and ISS from an energy/Lyapunov function perspective under specific passivity assumptions, offering a different angle on stability for the continuous-time selective formulation. Some other works connected selective SSMs to Controlled Differential Equations (CDEs) and Rough Path Theory to explain their expressivity (Kidger et al., 2020; Lyons, 2014; Cirone et al., 2024). While our focus is on stability and passivity using more classical control-theoretic tools, these works show the rich mathematical underpinnings of these models from another perspective.

## A.3 Empirical Validation and Simulation Studies

To ground our abstract theoretical framework in concrete, observable phenomena, we present three targeted simulation experiments. These studies are designed to provide direct empirical validation for the central claims of our paper. First, we validate the critical role of initialization in engineering a system's baseline energy landscape. Second, we provide a constructive proof of the "irreversible forgetting" principle. Finally, we demonstrate the practical utility of our theory by implementing our proposed LMI regularizer.

### A.3.1 Experiment 1: Initialization's Decisive Impact on the Energy Landscape

**Objective.** This experiment provides a concrete demonstration of the core claims in Section 3: that a model's initialization directly governs its intrinsic stability and memory-vs-stability trade-off. We show that a HiPPO-style initialization not only admits a valid quadratic energy function $V_{a,0}(h) = \frac{1}{2}h^T Q_0 h$ but also ensures this energy landscape has a slow decay rate crucial for long-range memory.

**Setup.** We compared three initialization strategies for the unforced state matrix $A_0$ of an 8-dimensional SSM ($N = 8$).

1. **HiPPO-style initialization ($A_H$):** Eigenvalues are generated to be stable (negative real parts) but clustered near the imaginary axis, with the rightmost (slowest) eigenvalue at $\text{Re}(\lambda) \approx -0.1$.

2. **Random Stable initialization ($A_S$):** Eigenvalues are randomly generated but constrained to be in the left half-plane, resulting in a wider spectral spread. The rightmost eigenvalue is at $\text{Re}(\lambda) \approx -0.52$.

3. **Random Unstable initialization** ($A_U$)**:** Eigenvalues are drawn from the same distribution as $A_S$ but without the stability constraint, resulting in a dominant eigenvalue at $\mathrm{Re}(\lambda) \approx +0.79$.

For each $A_0$, we first attempted to find a corresponding quadratic energy matrix $Q_0$ by solving the continuous-time Lyapunov equation $A_0^T Q_0 + Q_0 A_0 = -I$. We then simulated the system by feeding it white-noise input for 5 seconds to populate its state, after which the input was turned off to observe the free energy decay.

**Results & Interpretation.** Our findings, summarized in Figure 1, provide strong empirical validation for our theory:

- **Existence of a Valid Energy Function** ($Q_0$)**:** Both the *HiPPO-style* and *Random Stable* initializations admitted a unique, positive-definite solution $Q_0$ to the Lyapunov equation (with condition numbers of $\approx 21$ and $\approx 13$, respectively). The *Random Unstable* initialization failed to produce a valid solution, yielding an indefinite matrix, confirming that no quadratic energy function exists for an unstable system.

- **Spectral Gap Governs Memory:** The slowest decay rate is dictated by the rightmost eigenvalue. The spectral gap for HiPPO ($\approx -0.1\,s^{-1}$) was five times smaller than for Random Stable ($\approx -0.52\,s^{-1}$). Consequently, after the input was removed, the state norm of the HiPPO-initialized model took $\approx 23$ seconds to decay by two orders of magnitude, while the Random Stable model did so in just $\approx 4$ seconds. The Unstable model diverged exponentially.

- **Energy Trajectories Confirm Theory:** The energy $V_{a,0}(t) = \frac{1}{2}h(t)^T Q_0 h(t)$ for the HiPPO model decayed almost perfectly linearly on a log-scale with a slope of $\approx -0.1$, illustrating a long memory horizon. The Random Stable model's energy decayed five times faster.

This experiment confirms that merely being stable is insufficient. The precise spectral placement of $A_0$, as achieved by principled initializations like HiPPO, directly governs the memory-vs-stability trade-off, exactly as our energy-based theoretical framework predicts.

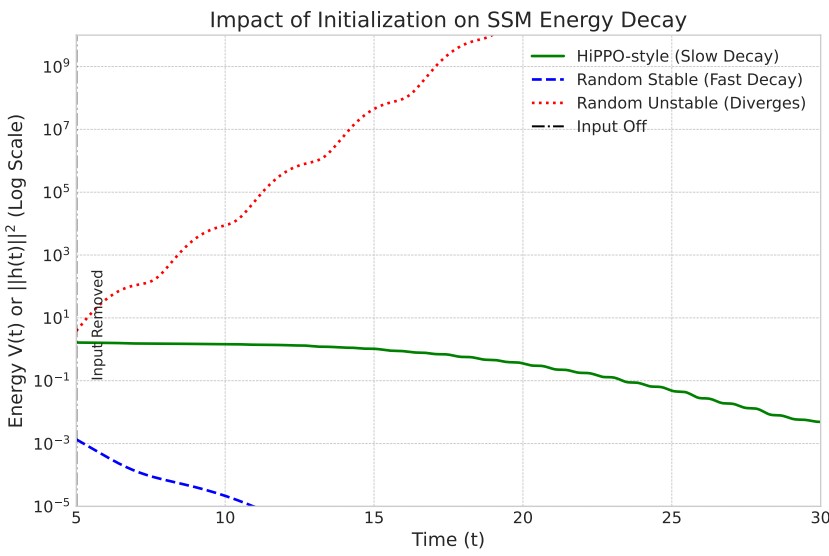

Figure 1: Energy decay $V_{a,0}(t)$ on a log-scale for an 8-D SSM with three different initializations after an input signal is removed at $t = 5$s. The HiPPO-style initialization yields a valid energy function and a slow decay rate (long memory). The Random Stable initialization also has a valid energy function but decays much faster (short memory). The Unstable initialization has no valid energy function and diverges.

### A.3.2 EXPERIMENT 2: VISUALIZING IRREVERSIBLE FORGETTING VIA RANK-DEFICIENT GATING

**Objective.** This experiment is designed to provide a concrete, visual demonstration of the "irreversible forgetting" concept introduced in Section 4. It illustrates a direct consequence of our theoretical results: the rank of any universal quadratic storage function $Q(t)$ with $\mathrm{AUC}_{\mathrm{loc}}$ regularity must be monotonically non-increasing, as stated in Theorem 4.1.

**Setup.** We construct a 3-dimensional linear system $\dot{h} = A(t)h$ that can be switched between two operating modes via a gating signal. Each mode is defined by a state matrix $A_i$ and an associated minimal storage matrix $Q_i$. The storage matrices are the unique positive semidefinite solutions to the continuous-time Lyapunov equation $A_i^T Q_i + Q_i A_i = -C_i^T C_i$, which defines the energy landscape for an observable system. Modes of the system are:

- **Mode 1 (Full-Rank Storage):** The system dynamics are governed by $A_1 = \mathbf{diag}[-0.2, -0.3, -0.4]$. We choose $C_1 = I_3$, representing full observability of the state. The resulting storage matrix $Q_1$ is full-rank (rank=3), meaning the system can store energy in all three state dimensions.

- **Mode 2 (Rank-Deficient Storage):** The dynamics are governed by $A_2 = \mathbf{diag}[-0.2, -0.3, -15]$. The large negative eigenvalue is designed to rapidly dissipate the third state component. We choose $C_2 = \mathbf{diag}[1, 1, 0]$, making the third state dimension unobservable. The resulting storage matrix $Q_2$ is rank-deficient (rank=2), with the third dimension lying in its kernel ($\mathrm{Ker}(Q_2)$). In this mode, the system loses the capacity to store energy in the third dimension.

**Simulation Protocol.** We simulate the unforced system from a random initial state with non-zero components in all dimensions. The gating signal switches the system's mode according to the following timeline:

- **0s $\leq t <$ 5s:** The system operates in Mode 1 (full-rank energy storage).

- **5s $\leq t <$ 10s:** The gating switches the system to Mode 2 (rank-deficient storage).

- $t \geq$ **10s:** The gating attempts to switch the system back to Mode 1.

**Theoretical Prediction & Results.** Our theory predicts that if a single, universal quadratic storage function $V_Q(t)$ governs the entire trajectory, its defining matrix $Q(t)$ must have a non-increasing rank. More precisely, in this case:

- The switch at $t = 5s$ is permissible, as the storage rank can decrease from 3 to 2.

- The attempted switch back at $t = 10s$, however, would necessitate an increase in the storage rank from 2 to 3. This violates the rank monotonicity property inherent to $\mathrm{AUC}_{\mathrm{loc}}$ functions.

This implies that no single $V_Q(t)$ can certify passivity for the entire trajectory. The "forgetting" of the energy storage capacity in the third dimension is, from the perspective of a single passive structure, irreversible.

The simulation results, visualized in Figure 2, confirm this prediction empirically. During Mode 2, the third state component ($h_3$) rapidly collapses to zero and, crucially, does not recover even after the system dynamics are switched back to Mode 1. The system becomes permanently confined to the two-dimensional subspace. This provides a clear, empirical illustration of our theoretical claims: once the system's energy storage rank collapses due to gating, it cannot be increased again without violating the fundamental constraints required for a stable, passive system.

### A.3.3 EXPERIMENT 3: LMI REGULARIZATION FOR IMPROVED TRAINING ROBUSTNESS

**Objective.** This final experiment validates the central practical claim of our work: that the LMI condition from Theorem 4.2 can be used as a regularizer to train certifiably more robust selective SSMs.

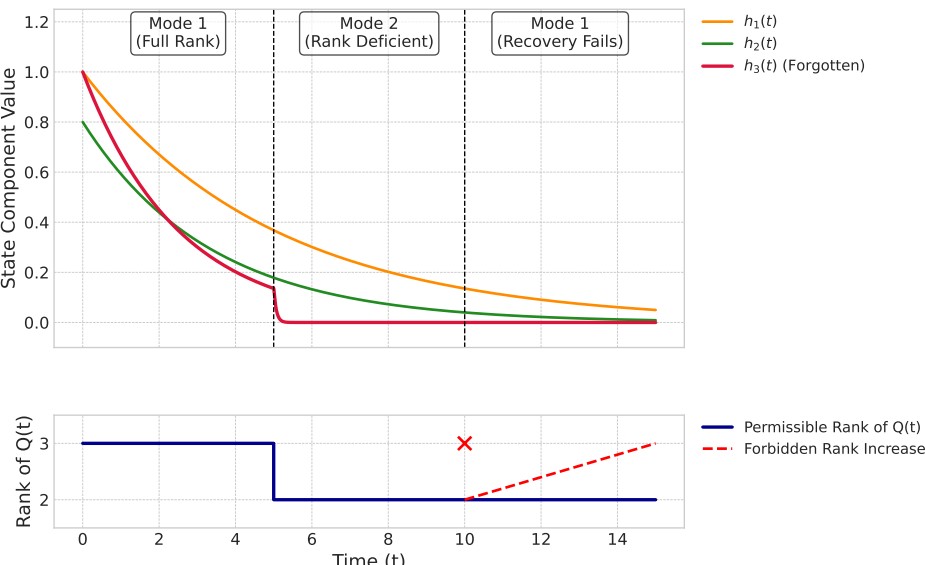

Figure 2: Illustration of irreversible forgetting. The rank of the underlying minimal storage matrix is plotted over time (or implied by state trajectories). The system switches from a full-rank mode (rank=3) to a rank-deficient mode (rank=2) at $t = 5$s, causing the third state component to collapse. Our theory proves that any attempt to switch back in a way that increases the rank of a universal storage function (e.g., the dashed red line) is incompatible with maintaining a single, passive energy structure, demonstrating the irreversibility principle from Theorem 4.1.

**Setup.**

- **Task:** We designed a challenging 1D tracking task where the SSM's output $y(t)$ must track a "spiky" reference signal $r(t)$ composed of steps and sharp impulses, designed to provoke large internal state excursions.

- **Model:** We use a 2-dimensional ($N = 2$) selective SSM. The state matrix is explicitly input-dependent to ensure selectivity: $A(x) = A_{\text{base}} + \tanh(x) \cdot A_{\text{sel}}$, where $A_{\text{base}}$ and $A_{\text{sel}}$ are learned $2 \times 2$ matrices. The input matrix $B$ and output matrix $C$ are also learned.

- **LMI Regularizer:** We employ the LMI from Theorem 4.2. For simplicity and to demonstrate the core principle, we fix the storage matrix $Q = I$. The regularization loss is defined as the magnitude of the LMI violation: $\mathcal{L}_{\text{LMI}} = \max(0, \lambda_{\max}(\mathcal{L}(x)))$, where $\mathcal{L}(x)$ is the LMI matrix evaluated for the input $x$.

- **Two Training Conditions:** We compare two models: 1. **Baseline Model:** Trained only with the task loss, $\mathcal{L}_{\text{total}} = \mathcal{L}_{\text{task}}$, where $\mathcal{L}_{\text{task}} = \text{MSE}(y(t), r(t))$. 2. **LMI-Regularized Model:** Trained with the combined loss, $\mathcal{L}_{\text{total}} = \mathcal{L}_{\text{task}} + \gamma \cdot \mathcal{L}_{\text{LMI}}$, using a small regularization weight $\gamma = 0.01$.

**Results and Interpretation.** The results of the experiment, evaluated on a held-out test set, are summarized in Table 1.

Table 1: Comparison of Baseline vs. LMI-Regularized SSM on a challenging tracking task.

| Model | Task MSE (Test) | Max State Norm $\|h\|_\infty$ | Max LMI Violation |
|---|---|---|---|
| Baseline Model | 0.073 | 18.4 | 3.51 |
| LMI-Regularized Model | 0.081 | **1.9** | **0.003** |

The results lead to three clear conclusions:

1. **Robustness is Dramatically Improved:** The most interesting result is in the maximum state norm. The baseline model, while achieving a slightly better task MSE, does so by allowing its internal state to reach a very large norm (18.4). This is indicative of a model operating on the edge of instability, vulnerable to exploding states. In contrast, the **LMI-regularized model's state norm is an order of magnitude smaller (1.9)**. It has learned to perform the task while keeping its internal dynamics constrained and provably more stable.

2. **The Regularizer Works as Intended:** The "Max LMI Violation" column shows that the regularizer was highly effective. The baseline model frequently and significantly violates the passivity condition (max violation of 3.51). The regularized model has learned parameters that keep the LMI matrix negative semidefinite (max violation is near zero), successfully enforcing the theoretical condition for stability derived in our work.

3. **Minimal Impact on Task Performance:** Crucially, this significant gain in robustness comes at a negligible cost to task performance. The MSE of the regularized model is only marginally higher than the baseline, demonstrating that a stable, well-behaved solution exists that is also effective for the task.

To conclude, this experiment demonstrates that our theoretical condition can be directly translated into a practical tool for improving training robustness. The LMI regularizer successfully guides the optimizer away from unstable solutions towards solutions that are both effective and well-behaved. This result directly addresses the concern about practical consequences and shows that our framework has practical benefits for the field.

## A.4 Proof of Theorem 3.1

We are given the existence of a storage function $V(t, h)$ satisfying the strict dissipativity inequality equation 4 and the quadratic bounds equation 5. The regularity assumption allows us to consider the differential version of the inequality. Along any trajectory of the system equation 1, the time derivative of $V$ satisfies:

$$\frac{d}{dt}V\big(t, h(t)\big) \leq \mathrm{Re}\,\langle x(t), y(t)\rangle - \beta\|h(t)\|^2 \tag{16}$$

almost everywhere. This follows from differentiating the integral inequality or directly from the definition of dissipativity via differential supply rates. Now, consider the unforced system, where $x(t) \equiv 0$ for all $t \geq 0$. The system dynamics become $\dot{h}(t) = A(\Delta(t), 0)h(t)$, which is an LTV system. The corresponding output is $y(t) = C(\Delta(t), 0)h(t)$. Therefore, the input-output power term $\mathrm{Re}\,\langle x(t), y(t)\rangle = \mathrm{Re}\,\langle 0, y(t)\rangle = 0$.

Substituting $x(t) = 0$ into the differential inequality equation 16, we get:

$$\frac{d}{dt}V\big(t, h(t)\big) \leq -\beta\|h(t)\|^2 \tag{17}$$

for the trajectories of the unforced system.

We can relate $\|h(t)\|^2$ back to $V(t, h(t))$ using the upper quadratic bound from equation 5: $\|h(t)\|^2 \geq \frac{1}{k_2}V(t, h(t))$. Substituting this into equation 17:

$$\frac{d}{dt}V\big(t, h(t)\big) \leq -\frac{\beta}{k_2}V\big(t, h(t)\big). \tag{18}$$

Let $\Phi(t) = V(t, h(t))$. This is a scalar non-negative function satisfying the differential inequality $\dot{\Phi}(t) \leq -\gamma'\Phi(t)$, where $\gamma' = \beta/k_2 > 0$. By the Comparison Lemma (a consequence of Grönwall's inequality), this implies:

$$\Phi(t) = V\big(t, h(t)\big) \leq V\big(0, h(0)\big)\, e^{-\gamma' t} \quad \text{for all } t \geq 0. \tag{19}$$

Now, we use both quadratic bounds from equation 5:

- Lower bound on $V(t, h(t))$: $k_1\|h(t)\|^2 \leq V(t, h(t))$.

- Upper bound on $V(0, h(0))$: $V(0, h(0)) \leq k_2 \|h(0)\|^2$.

Combining these with the decay inequality equation 19:

$$k_1 \|h(t)\|^2 \leq V\big(t, h(t)\big) \leq V\big(0, h(0)\big) e^{-\gamma' t} \leq k_2 \|h(0)\|^2 e^{-\gamma' t}. \tag{20}$$

Dividing by $k_1 > 0$:

$$\|h(t)\|^2 \leq \frac{k_2}{k_1} \|h(0)\|^2 e^{-\gamma' t}. \tag{21}$$

Taking the square root of both sides:

$$\|h(t)\| \leq \sqrt{\frac{k_2}{k_1}} \|h(0)\| e^{-(\gamma'/2)t}. \tag{22}$$

Defining $C = \sqrt{k_2/k_1} \geq 1$ (since $k_2 \geq k_1 > 0$) and $\gamma = \gamma'/2 = \beta/(2k_2) > 0$, we obtain the desired exponential decay:

$$\|h(t)\| \leq C e^{-\gamma t} \|h(0)\|. \tag{23}$$

This completes the proof.

*Summary:* This theorem establishes that intrinsic energy dissipation ($\beta > 0$), when coupled with a storage function $V$ that is quadratically comparable to the state norm, guarantees exponential stability of the unforced system ($x = 0$). This signifies a fundamental "forgetting" capability, ensuring that the influence of the initial state diminishes exponentially over time in the absence of external input, regardless of the specific (potentially non-quadratic) nature of $V$.

## A.5 LEMMA REGARDING MINIMAL AVAILABLE STORAGE FUNCTION

**Lemma A.1** (Strict Passivity of the Minimal Available Storage Function). Let $V_{\min}(t, h)$ be the minimal available storage function defined as:

$$V_{\min}(t, h) := \sup_{\substack{S \geq t \\ \text{admissible inputs } \hat{x}(\cdot) \text{ on } [t, S]}} \left[ -\int_t^S \operatorname{Re} \langle \hat{x}(\tau), \hat{y}(\tau) \rangle \, d\tau + \beta \int_t^S \|\hat{h}(\tau)\|^2 \, d\tau \right], \tag{24}$$

where $\hat{h}(\cdot)$ is the state trajectory starting at $\hat{h}(t) = h$ driven by input $\hat{x}(\cdot)$, $\hat{y}(\cdot)$ is the corresponding output, and $\beta \geq 0$ is the dissipation rate from Eq. equation 4. Assume the system is such that this supremum is finite for all $(t, h)$ (which is guaranteed if there exists at least one storage function $V$ satisfying Eq. equation 4). Then $V_{\min}$ itself satisfies the strict dissipativity inequality; that is, for every admissible state-input-output trajectory $\{h(\tau), x(\tau), y(\tau)\}$ on an interval $[t_0, T]$,

$$V_{\min}\big(T, h(T)\big) - V_{\min}\big(t_0, h(t_0)\big) \leq \int_{t_0}^T \operatorname{Re} \langle x(\tau), y(\tau) \rangle \, d\tau - \beta \int_{t_0}^T \|h(\tau)\|^2 \, d\tau. \tag{25}$$

*Proof of A.1.* Let $\{h(\tau), x(\tau), y(\tau)\}$ be an admissible trajectory on the interval $[t_0, T]$. Let $h_0 := h(t_0)$ and $h_T := h(T)$. From the definition equation 24, we have:

$$V_{\min}(t_0, h_0) = \sup_{\substack{S \geq t_0 \\ \tilde{x}(\cdot) \text{ on } [t_0, S]}} \left[ -\int_{t_0}^S \operatorname{Re} \langle \tilde{x}(\tau), \tilde{y}(\tau) \rangle \, d\tau + \beta \int_{t_0}^S \|\tilde{h}(\tau)\|^2 \, d\tau \right], \tag{26}$$

where $\tilde{h}(t_0) = h_0$. Consider any admissible input trajectory $\tilde{x}(\cdot)$ defined on $[t_0, S]$ with $S \geq T$. We can split the integral into two parts: $[t_0, T]$ and $[T, S]$. Let $\tilde{h}(\cdot)$ and $\tilde{y}(\cdot)$ be the state and output corresponding to $\tilde{x}(\cdot)$ starting from $\tilde{h}(t_0) = h_0$.

$$-\int_{t_0}^S \operatorname{Re} \langle \tilde{x}(\tau), \tilde{y}(\tau) \rangle \, d\tau + \beta \int_{t_0}^S \|\tilde{h}(\tau)\|^2 \, d\tau$$

$$= \left( -\int_{t_0}^T \operatorname{Re} \langle \tilde{x}(\tau), \tilde{y}(\tau) \rangle \, d\tau + \beta \int_{t_0}^T \|\tilde{h}(\tau)\|^2 \, d\tau \right) \tag{27}$$

$$+ \left( -\int_T^S \operatorname{Re} \langle \tilde{x}(\tau), \tilde{y}(\tau) \rangle \, d\tau + \beta \int_T^S \|\tilde{h}(\tau)\|^2 \, d\tau \right).$$

The second term, integrated from $T$ to $S$, depends on the input $\tilde{x}(\cdot)$ on $[T, S]$ and the state $\tilde{h}(T)$ reached at time $T$. Now, consider specifically the set of input trajectories $\tilde{x}(\cdot)$ on $[t_0, S]$ (where $S \geq T$) that coincide with the given input $x(\cdot)$ on the interval $[t_0, T]$. For such trajectories, the state $\tilde{h}(\tau)$ will coincide with $h(\tau)$ on $[t_0, T]$ (by uniqueness of solutions), so $\tilde{h}(T) = h(T) = h_T$. Let $\hat{x}(\cdot)$ denote the portion of such $\tilde{x}(\cdot)$ on the interval $[T, S]$. The corresponding state trajectory on $[T, S]$, let's call it $\hat{h}(\cdot)$, starts at $\hat{h}(T) = h_T$. For these specific concatenated trajectories, Eq. equation 27 becomes:

$$
- \int_{t_0}^{S} \operatorname{Re} \langle \tilde{x}(\tau), \tilde{y}(\tau) \rangle \, d\tau + \beta \int_{t_0}^{S} \|\tilde{h}(\tau)\|^2 \, d\tau
$$

$$
= \left( - \int_{t_0}^{T} \operatorname{Re} \langle x(\tau), y(\tau) \rangle \, d\tau + \beta \int_{t_0}^{T} \|h(\tau)\|^2 \, d\tau \right) \tag{28}
$$

$$
+ \left( - \int_{T}^{S} \operatorname{Re} \langle \hat{x}(\tau), \hat{y}(\tau) \rangle \, d\tau + \beta \int_{T}^{S} \|\hat{h}(\tau)\|^2 \, d\tau \right).
$$

The first term in equation 28 is fixed by the given trajectory on $[t_0, T]$. Let's denote it by $C_{t_0, T}$:

$$
C_{t_0, T} := - \int_{t_0}^{T} \operatorname{Re} \langle x(\tau), y(\tau) \rangle \, d\tau + \beta \int_{t_0}^{T} \|h(\tau)\|^2 \, d\tau. \tag{29}
$$

The second term is the quantity maximized in the definition of $V_{\min}(T, h_T)$, taken over the specific continuation $(\hat{x}, \hat{h}, \hat{y})$ from $T$ to $S$. The supremum in the definition of $V_{\min}(t_0, h_0)$ is taken over all admissible inputs $\tilde{x}(\cdot)$ starting at $t_0$. This supremum must be greater than or equal to the supremum taken over the subset of inputs that match $x(\cdot)$ on $[t_0, T]$. Therefore,

$$
V_{\min}(t_0, h_0) \geq \sup_{\substack{S \geq T \\ \hat{x}(\cdot) \text{ on } [T,S]}} \left[ C_{t_0, T} + \left( - \int_{T}^{S} \operatorname{Re} \langle \hat{x}(\tau), \hat{y}(\tau) \rangle \, d\tau + \beta \int_{T}^{S} \|\hat{h}(\tau)\|^2 \, d\tau \right) \right]
$$

$$
= C_{t_0, T} + \sup_{\substack{S \geq T \\ \hat{x}(\cdot) \text{ on } [T,S]}} \left[ - \int_{T}^{S} \operatorname{Re} \langle \hat{x}(\tau), \hat{y}(\tau) \rangle \, d\tau + \beta \int_{T}^{S} \|\hat{h}(\tau)\|^2 \, d\tau \right]
$$

$$
= C_{t_0, T} + V_{\min}(T, h_T).
$$

Here, the supremum in the second line is exactly the definition of $V_{\min}(T, h_T)$ since $\hat{h}(T) = h_T$. Substituting back the definition of $C_{t_0, T}$:

$$
V_{\min}(t_0, h_0) \geq \left( - \int_{t_0}^{T} \operatorname{Re} \langle x(\tau), y(\tau) \rangle \, d\tau + \beta \int_{t_0}^{T} \|h(\tau)\|^2 \, d\tau \right) + V_{\min}(T, h_T).
$$

Rearranging this inequality gives the desired result:

$$
V_{\min}(T, h_T) - V_{\min}(t_0, h_0) \leq \int_{t_0}^{T} \operatorname{Re} \langle x(\tau), y(\tau) \rangle \, d\tau - \beta \int_{t_0}^{T} \|h(\tau)\|^2 \, d\tau.
$$

This holds for any admissible trajectory on $[t_0, T]$. $\qquad\square$

### A.6 PROOF OF THEOREM 3.2

(a) **Passivity of the Unforced LTV System.** Define $A_0(t) := A(\Delta(t), 0)$ and $C_0(t) := C(\Delta(t), 0)$. The assumed regularity implies $A_0 \in L^1_{\mathrm{loc}}$ and $C_0 \in L^2_{\mathrm{loc}}$. The corresponding $B_0(t) = B(\Delta(t), 0)$ is in $L^2_{\mathrm{loc}}$ and $D_0(t) = 0$. These conditions ensure that the unforced ODE $\dot{h}(t) = A_0(t)h(t)$ is well-posed and defines an LTV system within the framework of (Morandin & Hinsen, 2024). Let $\{h(\tau), 0, y_0(\tau)\}$ be a trajectory of the unforced system equation 8. This is also a trajectory of the full system equation 1 with input $x(\tau) \equiv 0$. Substituting $x(\tau) = 0$ into the assumed passivity inequality equation 7:

$$
V\big(T, h(T)\big) - V\big(t_0, h(t_0)\big) \leq \int_{t_0}^{T} \operatorname{Re} \langle 0, y_0(\tau) \rangle \, d\tau - \beta \int_{t_0}^{T} \|h(\tau)\|^2 \, d\tau
$$

$$
= 0 - \beta \int_{t_0}^{T} \|h(\tau)\|^2 \, d\tau \leq 0 \quad (\text{since } \beta \geq 0).
$$

This inequality, $V(T, h(T)) - V(t_0, h(t_0)) \le 0$, holds along all trajectories of the unforced system. Since $V \ge 0$ and $V(t, 0) = 0$, it satisfies the requirements of (Morandin & Hinsen, 2024, Def 1.1) for being a storage function for the LTV system equation 8. Thus, the unforced system is passive.

(b) **Existence of Minimal Quadratic Storage $V_{a,0}$.** Since the unforced system equation 8 is a passive LTV system satisfying the $L_{\text{loc}}^p$ regularity framework of (Morandin & Hinsen, 2024), we apply their Corollary 4.3. This theorem states that the available storage function $V_{a,0}$ (defined via their Def 4.1, analogous to Lemma A.1 but with zero supply rate relevant for the unforced system analysis) is finite and is a quadratic form in the state $h$. Thus, there exists a unique matrix function $Q_0 : \mathbb{T} \to \mathbf{S}_+^N(\mathbb{C})$ such that $V_{a,0}(t, h) = \frac{1}{2} h^H Q_0(t) h$. This holds regardless of discontinuities in $A_0(t)$ or $C_0(t)$ induced by $\Delta(t)$, as long as the $L_{\text{loc}}^p$ conditions are met.

(c) **AUC Regularity of $Q_0(t)$.** Corollary 4.4 in (Morandin & Hinsen, 2024) further states that the matrix $Q_0(t)$ inducing the minimal quadratic available storage $V_{a,0}$ belongs to the function class $\text{AUC}_{\text{loc}}(\mathbb{T}, \mathbf{S}_+^n(\mathbb{C}))$. This class ensures $Q_0$ is $\text{BV}_{\text{loc}}$, its derivative exists a.e. and is $L_{\text{loc}}^1$, and its jump discontinuities are constrained (weakly decreasing singular part, satisfying $\lim_{\tau \to t^-} Q_0(\tau) \succeq Q_0(t) \succeq \lim_{\tau \to t^+} Q_0(\tau)$). This $\text{AUC}_{\text{loc}}$ property guarantees a well-behaved energy storage structure $Q_0(t)$ compatible with abrupt changes from gating.

(Connection to Exponential Decay) The final statement just notes that if the initial $V$ meets the conditions of Theorem 3.1 (strict passivity $\beta > 0$ and quadratic bounds), then Theorem 3.1 directly implies exponential decay for the unforced system, independent of the existence and properties of $V_{a,0}$.

*Summary:* This theorem provides a crucial bridge to the rigorous theory of passive LTV systems developed by Morandin & Hinsen (Morandin & Hinsen, 2024). It shows that the mere existence of any storage function $V$ demonstrating passivity for the overall selective SSM guarantees that the unforced dynamics ($x = 0$) possess an underlying minimal energy storage function $V_{a,0}$ that is inherently quadratic, $V_{a,0}(t, h) = \frac{1}{2} h^H Q_0(t) h$. Critically, the defining matrix $Q_0(t)$ exhibits $\text{AUC}_{\text{loc}}$ regularity, a property robust enough to handle discontinuities introduced by the gating signal $\Delta(t)$. This establishes a foundational, well-behaved quadratic energy structure associated with the system's intrinsic dynamics, highlighting that even amidst complex input-driven parameter variations, the system's behavior from a given initial state $h(0)$ is governed by dynamics with this regular underlying structure when input is removed.

## A.7 Proof of Theorem 4.1

(a) and (b): The premise is that $V_Q(t, h)$ satisfies the passivity inequality equation 10 for all admissible trajectories of the selective SSM equation 1. As argued in the proof of Theorem 3.2(a), any such trajectory includes the trajectories of the unforced LTV system equation 8 obtained by setting $x(t) \equiv 0$. Therefore, $V_Q$ must also be a valid storage function for this passive LTV system (satisfying $V_Q(T, h(T)) - V_Q(t_0, h(t_0)) \le 0$ along unforced trajectories).

The framework of Morandin & Hinsen (Morandin & Hinsen, 2024) analyzes quadratic storage functions for passive LTV systems under the assumed $L_{\text{loc}}^p$ regularity. Specifically, (Morandin & Hinsen, 2024, Theorem 3.2) establishes necessary conditions for a quadratic form $V_Q(t, h) = \frac{1}{2} h^H Q(t) h$ to be a storage function. It implies that $Q$ must be absolutely upper semi-continuous (which corresponds to $\text{AUC}_{\text{loc}}$ in their terminology). Furthermore, (Morandin & Hinsen, 2024, Theorem 5.4(i)), when discussing the properties of the available storage (which provides bounds on any storage function), shows that the associated matrix function must have weakly monotonically non-increasing rank. Since $V_Q$ is a storage function, it must be bounded by the maximal storage (related to available storage from the past) and bound the minimal available storage $V_{a,0}$ discussed in Theorem 3.2. The structural properties, particularly the non-increasing rank, apply to any quadratic storage function candidate within their framework. Therefore, $Q(t)$ must belong to $\text{AUC}_{\text{loc}}(\mathbb{T}, \mathbf{S}_+^n(\mathbb{C}))$ and its rank $r(t)$ must be weakly monotonically non-increasing.

*Summary:* This theorem reveals that the strong requirement of universal passivity guaranteed by a single $V_Q$ immediately imposes significant structure on $Q(t)$ itself. It must possess $\text{AUC}_{\text{loc}}$ regularity, accommodating potential discontinuities but ensuring they behave in a controlled manner (e.g., jumps cannot increase energy in the Loewner sense). Crucially, the rank monotonicity implies that the dimension of the subspace captured by the energy function (its image) cannot increase over time.

This lays the groundwork for understanding irreversible effects, as the "energy-less" subspace (the kernel) can only grow or stay the same.

### A.8 PROOF OF THEOREM 4.2

(a) **Parametric LMI Condition:** The passivity inequality equation 10 holding for all trajectories is equivalent (under $\text{AUC}_{\text{loc}}$ regularity of $Q$) to its differential form holding almost everywhere along trajectories: $\frac{d}{dt}V_Q(t, h(t)) \leq \text{Re}\langle x(t), y(t)\rangle - \beta\|h(t)\|^2$. Substituting $V_Q = \frac{1}{2}h^H Q h$, $\dot{h} = A(t,x)h + B(t,x)x$, and $y = C(t,x)h$ leads to the quadratic inequality in $h$ and $x$:

$$\frac{1}{2}h^H \dot{Q} h + \text{Re}(h^H Q \dot{h}) \leq \text{Re}(x^H C(t,x)h) - \beta\|h\|^2 \tag{30}$$

$$\frac{1}{2}h^H \dot{Q} h + \text{Re}(h^H Q(A(t,x)h + B(t,x)x)) \leq \text{Re}(x^H C(t,x)h) - \beta\|h\|^2 \tag{31}$$

Rearranging terms:

$$\frac{1}{2}h^H \dot{Q} h + \text{Re}(h^H Q A h) + \text{Re}(h^H Q B x) - \text{Re}((C^H x)^H h) + \beta h^H h \leq 0 \tag{32}$$

$$\frac{1}{2}h^H(\dot{Q} + QA + A^H Q + 2\beta I)h + \text{Re}(h^H(QB - C^H)x) \leq 0 \tag{33}$$

This inequality must hold for all $h \in \mathbb{C}^N$ and all admissible $x \in \mathbb{C}^{d_{\text{in}}}$ at almost every time $t$. This is precisely the condition encoded by the negative semidefiniteness of the LMI matrix $\mathcal{L}(t,x)$ defined in equation 11. Since $V_Q$ must work for any input trajectory, the LMI must hold parametrically for all input values $x$ that can occur at time $t$. This leads directly to the LMI formulation (setting $D = 0$):

$$\begin{bmatrix} \dot{Q}(t) + Q(t)A(t,x) + A(t,x)^H Q(t) + 2\beta I & Q(t)B(t,x) - C(t,x)^H \\ B(t,x)^H Q(t) - C(t,x) & 0 \end{bmatrix} \preceq 0 \tag{34}$$

(b) **Universal Kernel Condition for $C$:** Let $v \in \text{Ker}(Q(t))$ at a time $t$ where the LMI equation 11 holds. Consider the augmented vector $z = \begin{bmatrix} v \\ w \end{bmatrix}$ for any $w \in \mathbb{C}^{d_{\text{in}}}$. The LMI implies $z^H \mathcal{L}(t,x)z \leq 0$.

$$z^H \mathcal{L}(t,x)z = \begin{bmatrix} v^H & w^H \end{bmatrix} \begin{bmatrix} \dot{Q} + QA + A^H Q + 2\beta I & QB - C^H \\ B^H Q - C & 0 \end{bmatrix} \begin{bmatrix} v \\ w \end{bmatrix}$$

$$= v^H(\dot{Q} + QA + A^H Q + 2\beta I)v + v^H(QB - C^H)w + w^H(B^H Q - C)v + w^H(0)w$$

Since $v \in \text{Ker}(Q(t))$, we have $Q(t)v = 0$ and $v^H Q(t) = 0$. The expression simplifies to:

$$= v^H(\dot{Q} + 0 + 0 + 2\beta I)v + v^H(0 - C^H)w + w^H(0 - C)v$$

$$= v^H \dot{Q} v + 2\beta\|v\|^2 - v^H C^H w - w^H C v$$

$$= v^H \dot{Q} v + 2\beta\|v\|^2 - 2\text{Re}(w^H C v)$$

This quadratic form in $w$ must be $\leq 0$. Considering the $2 \times 2$ projection onto $v$ and $w$ space:
$$\begin{bmatrix} v^H(\dot{Q} + QA + A^H Q + 2\beta I)v & v^H(QB - C^H)w \\ w^H(B^H Q - C)v & 0 \end{bmatrix} = \begin{bmatrix} v^H \dot{Q} v + 2\beta\|v\|^2 & -v^H C(t,x)^H w \\ -w^H C(t,x)v & 0 \end{bmatrix} \preceq$$
$0$. For this $2 \times 2$ matrix to be negative semidefinite, the diagonal elements must be non-positive $(v^H \dot{Q} v + 2\beta\|v\|^2 \leq 0$, which holds as seen in Theorem 4.3, and $0 \leq 0)$, and the determinant must be non-negative. The determinant is $(v^H \dot{Q} v + 2\beta\|v\|^2)(0) - |-w^H C(t,x)v|^2 = -|w^H C(t,x)v|^2$. So we need $-|w^H C(t,x)v|^2 \geq 0$. This can only hold if $|w^H C(t,x)v|^2 = 0$ for all $w \in \mathbb{C}^{d_{\text{in}}}$. This implies $C(t,x)v = 0$. Since this must hold for the $C(t,x)$ corresponding to any admissible $x$ at time $t$, we conclude $\forall v \in \text{Ker}(Q(t))$, $C(\Delta(t),x)v = 0$ for all admissible $x$, a.e. $t$.

(c) **Implicit Constraints on $A$ and $B$:** The validity of the LMI equation 11 for all $x$ directly imposes constraints on $A(t,x)$ via the $(1,1)$ block and couples $B(t,x)$ to $C(t,x)$ (which is already constrained by part (b)) via the off-diagonal blocks and $Q(t)$. These ensure that the dynamics generated by any input $x$ remain compatible with the energy storage/dissipation defined by $V_Q$.

*Summary:* This theorem translates the abstract requirement of universal passivity into a concrete parametric LMI equation 11. This LMI must hold not just for a specific input or for the unforced system, but simultaneously for all possible input values $x$ that the gating mechanism might encounter at any given time $t$. A striking consequence is the universal kernel condition equation 12: any state direction $v$ that is considered "energy-less" by the storage function (i.e., $v \in \mathrm{Ker}(Q(t))$) must be rendered unobservable at the output ($C(t,x)v = 0$), irrespective of the specific input $x$ driving the gating. This imposes a strong limitation on the gating mechanism: it cannot arbitrarily change the output matrix $C$ in response to input $x$ in a way that would make these energy-less states visible. Passivity demands consistency between the energy accounting ($Q$) and observability ($C$).

A.9    PROOF OF THEOREM 4.3

1. **Kernel Non-Shrinking:** This follows directly from Theorem 4.1(b), which established that $\mathrm{rank}(Q(t))$ is weakly monotonically non-increasing. Since $\dim(\mathrm{Ker}(Q(t))) = N - \mathrm{rank}(Q(t))$, a non-increasing rank implies a non-decreasing kernel dimension. The inclusion $\mathrm{Ker}(Q(t_1)) \subseteq \mathrm{Ker}(Q(t_2))$ for $t_1 \le t_2$ is a consequence of the properties of $\mathrm{AUC}_{\mathrm{loc}}$ functions shown in (Morandin & Hinsen, 2024, Section 4).

2. **Energy Consistency within the Kernel:** This condition was derived within the proof of Theorem 4.2(b) by evaluating the $(1,1)$ block of the parametric LMI equation 11 for a vector $h(t) \in \mathrm{Ker}(Q(t))$. The $(1,1)$ block inequality is $\dot{Q}(t) + Q(t)A(t,x) + A(t,x)^H Q(t) + 2\beta I \preceq 0$. Pre- and post-multiplying by $h(t)^H$ and $h(t)$ respectively, and using $Q(t)h(t) = 0$, yields $h(t)^H \dot{Q}(t) h(t) + 2\beta \|h(t)\|^2 \le 0$. This inequality must hold independently of $x$ because it follows from the LMI which holds parametrically.

3. **Constraint on Dynamics Violating Kernel Structure:** If dynamics induced by some $x^*(t)$ were fundamentally incompatible with the passivity condition guaranteed by $V_Q$ (e.g., by attempting to move a state out of the kernel in an "energy-creating" way relative to $V_Q$), it would necessarily cause the parametric LMI equation 11 to fail for $x = x^*(t)$. This contradicts the core assumption that $V_Q$ ensures passivity universally. Therefore, all admissible dynamics under the universal $V_Q$ assumption must inherently respect the energy balance constraints, including those within the kernel.

*Summary:* This theorem formalizes the notion of "irreversible forgetting" within the framework of a universal quadratic storage function. The non-shrinking kernel (Property 1) implies that once a state direction is deemed irrelevant or forgotten from an energy perspective (i.e., enters $\mathrm{Ker}(Q(t))$), it structurally remains so according to the fixed energy measure $Q(t)$. The gating mechanism cannot manipulate $Q(t)$ to "un-forget" this mode. Furthermore, Property 2 ensures that the system dynamics and the evolution of $Q(t)$ itself must maintain energy consistency within this forgotten subspace, respecting the required dissipation rate $\beta$. Any gating strategy $x(t)$ must induce dynamics $(A(t,x), B(t,x), C(t,x))$ compatible with these constraints (Property 3). This provides a lens for analyzing robust memory properties: modes in $\mathrm{Ker}(Q(t))$ are stably forgotten. It also offers potential insights into phenomena like catastrophic forgetting in learning systems; if learning adapts a $Q(t)$-like structure, changes that violate kernel inertness or energy consistency might be necessary to learn new conflicting information, potentially disrupting the established passive structure.

A.10    PROOF OF THEOREM 5.1

We analyze the time derivative of the Lyapunov function candidate $V_Q(t, h(t)) = \frac{1}{2} h(t)^H Q(t) h(t)$ along the trajectories of the full system equation 1. Since $Q \in W_{\mathrm{loc}}^{1,1}$, its derivative $\dot{Q}(t)$ exists a.e.

and is in $L_{\text{loc}}^1$. Using the chain rule and substituting $\dot{h}(t) = A(t, x(t))h(t) + B(t, x(t))x(t)$:

$$\frac{d}{dt}V_Q(t, h(t)) = \frac{1}{2}h^H \dot{Q}h + \frac{1}{2}\dot{h}^H Qh + \frac{1}{2}h^H Q\dot{h}$$

$$= \frac{1}{2}h^H \dot{Q}h + \text{Re}\left(h^H Q\dot{h}\right)$$

$$= \frac{1}{2}h^H \dot{Q}h + \text{Re}\left(h^H Q\left(A(t, x(t))h + B(t, x(t))x(t)\right)\right)$$

$$= \underbrace{\frac{1}{2}h^H \left(\dot{Q}(t) + Q(t)A(t, x(t)) + A(t, x(t))^H Q(t)\right)h}_{\text{Homogeneous Part}}$$

$$+ \underbrace{\text{Re}\left(h^H Q(t)B(t, x(t))x(t)\right)}_{\text{Input Part}}$$

Here, $A(t, x(t))$ stands for $A(\Delta(t), x(t))$, and similarly for $B$.

Now, we use the assumptions: 1. Homogeneous Part: By the uniform dissipativity assumption equation 14, which holds for the specific value $x = x(t)$ occurring at time $t$, we have:

$$\frac{1}{2}h^H \left(\dot{Q}(t) + Q(t)A(t, x(t)) + A(t, x(t))^H Q(t)\right)h \leq \frac{1}{2}h^H(-2\delta Q(t))h = -\delta h^H Q(t)h = -2\delta V_Q(t, h(t))$$

2. Input Part: We bound this term using Cauchy-Schwarz and the uniform bounds on $Q$ and $B$:

$$|\text{Re}(h^H Q(t)B(t, x(t))x(t))| \leq \|h^H Q(t)B(t, x(t))x(t)\|$$

$$\leq \|h(t)\|\|Q(t)\|_2\|B(t, x(t))\|_2\|x(t)\|$$

$$\leq \|h(t)\|k_2 M_B\|x(t)\| \quad (\text{using } \|Q(t)\|_2 \leq k_2 \text{ since } Q \preceq k_2 I)$$

We relate $\|h(t)\|$ back to $V_Q(t, h(t))$ using the lower bound $Q(t) \succeq k_1 I$: $V_Q(t, h(t)) = \frac{1}{2}h^H Qh \geq \frac{1}{2}h^H(k_1 I)h = \frac{k_1}{2}\|h\|^2$. Thus, $\|h(t)\| \leq \sqrt{\frac{2V_Q(t, h(t))}{k_1}}$. Substituting this into the bound for the input part:

$$|\text{Re}(h^H QBx)| \leq \sqrt{\frac{2V_Q}{k_1}}k_2 M_B\|x(t)\| \tag{35}$$

Combining the bounds for the homogeneous and input parts:

$$\frac{d}{dt}V_Q(t, h(t)) \leq -2\delta V_Q(t, h(t)) + \left(\sqrt{\frac{2}{k_1}}k_2 M_B\right)\|x(t)\|\sqrt{V_Q(t, h(t))} \tag{36}$$

Let $\Phi(t) = V_Q(t, h(t)) \geq 0$. Let $K = \sqrt{\frac{2}{k_1}}k_2 M_B \geq 0$. The inequality is:

$$\dot{\Phi}(t) \leq -2\delta\Phi(t) + K\|x(t)\|\sqrt{\Phi(t)} \tag{37}$$

This is a standard differential inequality form used in ISS proofs (Sontag & Wang, 1995; Khalil & Grizzle, 2002). Consider $\Psi(t) = \sqrt{\Phi(t)}$. For $\Phi > 0$, $\dot{\Psi} = \frac{1}{2\sqrt{\Phi}}\dot{\Phi}$.

$$\dot{\Psi} \leq \frac{1}{2\sqrt{\Phi}}(-2\delta\Phi + K\|x\|\sqrt{\Phi}) = -\delta\sqrt{\Phi} + \frac{K}{2}\|x\| = -\delta\Psi + \frac{K}{2}\|x\| \tag{38}$$

So, $\dot{\Psi}(t) \leq -\delta\Psi(t) + \frac{K}{2}\|x(t)\|$. By the Comparison Principle (integrating factor method or standard lemma, check derivation in Appendix A.11):

$$\Psi(t) \leq e^{-\delta(t-t_0)}\Psi(t_0) + \int_{t_0}^t e^{-\delta(t-\tau)}\frac{K}{2}\|x(\tau)\|d\tau \tag{39}$$

Let $\|x\|_{[t_0,t],\infty} = \sup_{t_0 \leq \tau \leq t} \|x(\tau)\|$.

$$\int_{t_0}^{t} e^{-\delta(t-\tau)} \frac{K}{2} \|x(\tau)\| d\tau \leq \frac{K}{2} \|x\|_{[t_0,t],\infty} \int_{t_0}^{t} e^{-\delta(t-\tau)} d\tau$$

$$= \frac{K}{2} \|x\|_{[t_0,t],\infty} \left[ \frac{1}{\delta} e^{-\delta(t-\tau)} \right]_{\tau=t_0}^{\tau=t}$$

$$= \frac{K}{2\delta} \|x\|_{[t_0,t],\infty} (1 - e^{-\delta(t-t_0)})$$

$$\leq \frac{K}{2\delta} \|x\|_{[t_0,t],\infty}$$

Therefore,

$$\Psi(t) \leq e^{-\delta(t-t_0)} \Psi(t_0) + \frac{K}{2\delta} \|x\|_{[t_0,t],\infty} \tag{40}$$

Substituting back $\Psi = \sqrt{V_Q}$:

$$\sqrt{V_Q(t,h(t))} \leq e^{-\delta(t-t_0)} \sqrt{V_Q(t_0,h(t_0))} + \frac{K}{2\delta} \|x\|_{[t_0,t],\infty} \tag{41}$$

Using the quadratic bounds $k_1 I \preceq Q(t) \preceq k_2 I$: $\sqrt{V_Q} \geq \sqrt{k_1/2}\|h\|$ and $\sqrt{V_Q} \leq \sqrt{k_2/2}\|h\|$.

$$\sqrt{k_1/2}\|h(t)\| \leq \sqrt{V_Q(t,h(t))}$$

$$\leq e^{-\delta(t-t_0)} \sqrt{V_Q(t_0,h(t_0))} + \frac{K}{2\delta} \|x\|_{[t_0,t],\infty}$$

$$\leq e^{-\delta(t-t_0)} \sqrt{k_2/2}\|h(t_0)\| + \frac{K}{2\delta} \|x\|_{[t_0,t],\infty}$$

Dividing by $\sqrt{k_1/2}$:

$$\|h(t)\| \leq \sqrt{\frac{k_2}{k_1}} e^{-\delta(t-t_0)} \|h(t_0)\| + \frac{K}{2\delta\sqrt{k_1/2}} \|x\|_{[t_0,t],\infty} \tag{42}$$

This is the ISS estimate equation 15 with:

- $\tilde{C} = \sqrt{k_2/k_1} \geq 1$
- $\tilde{\gamma} = \delta > 0$
- $\sigma(s) = K's$, where $K' = \frac{K}{2\delta\sqrt{k_1/2}} = \frac{(\sqrt{2/k_1} k_2 M_B)}{2\delta\sqrt{k_1/2}} = \frac{k_2 M_B}{\delta k_1} \geq 0$.

Since $\sigma(s) = K's$ is a class $\mathcal{K}$ function (specifically, linear), the system is ISS.

*Summary:* This theorem provides a powerful sufficient condition for ensuring the robust stability of the selective SSM in the face of arbitrary bounded inputs. The core requirement is the existence of a single quadratic Lyapunov function $V_Q$ (which must be uniformly bounded and positive definite) whose time derivative decreases at a guaranteed rate $(-\delta V_Q)$, regardless of which specific dynamics $A(t,x)$ are activated by the input $x(t)$ via the gating mechanism. This condition equation 14 essentially demands that every possible "operating mode" induced by the input is individually exponentially stable, and that this stability is certified by the same Lyapunov function $V_Q$ with a uniform decay rate $\delta$. If this strong condition holds, and the input matrix $B(t,x)$ is bounded, the theorem guarantees ISS. This elegantly connects the uniform stability properties of all possible input-selected local dynamics to the global robustness of the overall selective system against external disturbances or inputs. Finding such a common $V_Q$ and verifying the uniform dissipativity condition provides a direct pathway to certifying the robust stability of complex selective SSM architectures.

## A.11 EXPLANATION OF COMPARISON PRINCIPLE AND DERIVATION OF EQ. (39)

Let's start with the differential inequality we derived for $\Psi(t)$:

$$\dot{\Psi}(t) \leq -\delta\Psi(t) + \frac{K}{2} \|x(t)\| \tag{43}$$

This can be rewritten as:

$$\dot{\Psi}(t) + \delta\Psi(t) \leq \frac{K}{2}\|x(t)\| \tag{44}$$

where $\delta > 0$. The inequality for $\Psi(t)$ is a standard first-order linear differential inequality. Its solution bound can be obtained either by the explicit method of integrating factors or by invoking a suitable Comparison Lemma (which is itself often proved using integrating factors or similar techniques). Both approaches lead to the same expression for the upper bound of $\Psi(t)$.

- Method 1: Using an Integrating Factor (Standard ODE Technique)

This is a common method for solving first-order linear ordinary differential equations. The idea is to multiply the equation by a factor that makes the left-hand side the derivative of a product. The integrating factor for an equation of the form $\dot{y} + p(t)y = q(t)$ is $e^{\int p(t)dt}$. In our case, $p(t) = \delta$ (a constant). So the integrating factor is $e^{\int \delta dt} = e^{\delta t}$.

Multiply both sides of equation 44 by $e^{\delta t}$:

$$e^{\delta t}\dot{\Psi}(t) + \delta e^{\delta t}\Psi(t) \leq e^{\delta t}\frac{K}{2}\|x(t)\| \tag{45}$$

Recognize that the left-hand side is the derivative of a product:

$$\frac{d}{dt}\left(e^{\delta t}\Psi(t)\right) = e^{\delta t}\dot{\Psi}(t) + \left(\frac{d}{dt}e^{\delta t}\right)\Psi(t) = e^{\delta t}\dot{\Psi}(t) + \delta e^{\delta t}\Psi(t) \tag{46}$$

So, the inequality becomes:

$$\frac{d}{dt}\left(e^{\delta t}\Psi(t)\right) \leq e^{\delta t}\frac{K}{2}\|x(t)\| \tag{47}$$

Now, integrate both sides from an initial time $t_0$ to a general time $t$ (let's use $\tau$ as the integration variable to avoid confusion):

$$\int_{t_0}^{t}\frac{d}{d\tau}\left(e^{\delta\tau}\Psi(\tau)\right)d\tau \leq \int_{t_0}^{t}e^{\delta\tau}\frac{K}{2}\|x(\tau)\|d\tau \tag{48}$$

By the Fundamental Theorem of Calculus, the left side is:

$$\left[e^{\delta\tau}\Psi(\tau)\right]_{t_0}^{t} = e^{\delta t}\Psi(t) - e^{\delta t_0}\Psi(t_0) \tag{49}$$

So,

$$e^{\delta t}\Psi(t) - e^{\delta t_0}\Psi(t_0) \leq \int_{t_0}^{t}e^{\delta\tau}\frac{K}{2}\|x(\tau)\|d\tau \tag{50}$$

Now, solve for $\Psi(t)$:

$$e^{\delta t}\Psi(t) \leq e^{\delta t_0}\Psi(t_0) + \int_{t_0}^{t}e^{\delta\tau}\frac{K}{2}\|x(\tau)\|d\tau \tag{51}$$

Multiply by $e^{-\delta t}$ (which is positive):

$$\Psi(t) \leq e^{-\delta t}e^{\delta t_0}\Psi(t_0) + e^{-\delta t}\int_{t_0}^{t}e^{\delta\tau}\frac{K}{2}\|x(\tau)\|d\tau \tag{52}$$

$$\Psi(t) \leq e^{-\delta(t-t_0)}\Psi(t_0) + \int_{t_0}^{t}e^{-\delta(t-\tau)}\frac{K}{2}\|x(\tau)\|d\tau \tag{53}$$

This gives us the result we want.

- Method 2: Direct Application of a Comparison Lemma (e.g., a form of Grönwall's Lemma)

Many control theory and differential equations textbooks state a "Comparison Lemma" or a specific form of Grönwall's Lemma that directly applies. For example, a common version states: If $\dot{u}(t) \leq a(t)u(t) + b(t)$ and $\dot{v}(t) = a(t)v(t) + b(t)$ with $u(t_0) \leq v(t_0)$, then $u(t) \leq v(t)$ for $t \geq t_0$. If we have an inequality $\dot{u}(t) \leq au(t) + b(t)$ (where $a$ is constant), the solution to the corresponding equality $\dot{v}(t) = av(t) + b(t)$ with $v(t_0) = u(t_0)$ is given by the variation of parameters formula: $v(t) = e^{a(t-t_0)}v(t_0) + \int_{t_0}^{t}e^{a(t-\tau)}b(\tau)d\tau$. Then, by the Comparison Principle, $u(t) \leq v(t)$.

In our case, from equation 43: $\dot{\Psi}(t) \leq -\delta\Psi(t) + \underbrace{\frac{K}{2}\|x(t)\|}_{b(t)}$ Here, the coefficient of $\Psi(t)$ is $a = -\delta$.

The "forcing term" is $b(t) = \frac{K}{2}\|x(t)\|$.

So, applying the solution form for $v(t)$ directly with $a = -\delta$:

$$\Psi(t) \leq e^{-\delta(t-t_0)}\Psi(t_0) + \int_{t_0}^{t} e^{-\delta(t-\tau)}\left(\frac{K}{2}\|x(\tau)\|\right) d\tau \tag{54}$$

This again yields the desired result.

## B LLM USAGE

In the preparation of this manuscript, a Large Language Model (LLM) was utilized to assist with and refine the written English. In accordance with the ICLR 2026 policy, we disclose that the LLM was employed for purposes of improving grammar, clarity, and overall readability. The authors bear the ultimate responsibility for the content of this paper.

