# OpenReview forum: "Regularity and Stability Properties of Selective SSMs with Discontinuous Gating"
_ICLR.cc/2026/Conference — ICLR 2026 Conference Withdrawn Submission_

### Official Review · Reviewer_nz1P · 2025-10-21

**Soundness:** 3
**Presentation:** 3
**Contribution:** 3
**Rating:** 4
**Confidence:** 3

**Summary:**

This paper investigates the stability and regularity properties of continuous-time selective state-space models, a class of deep models whose parameters depend on the input and can change abruptly through discontinuous gating. The study draws on concepts from passivity theory and input-to-state stability to build a rigorous control-theoretic foundation for understanding such systems.

The authors first show that intrinsic energy dissipation ensures exponential forgetting of past states, even when the gating signals are discontinuous. They then prove that any passive selective state-space model possesses a minimal quadratic energy function whose defining matrix satisfies a form of absolute upper semicontinuity, guaranteeing stable energy behavior across switching dynamics. Building on this, the paper derives parametric linear matrix inequality conditions and kernel constraints that limit how gating mechanisms can influence the system while preserving passivity, formalizing a notion of irreversible forgetting. Finally, it provides sufficient conditions for global input-to-state stability, showing that uniform dissipativity leads to robustness under general inputs.

**Strengths:**

This paper presents a highly original and rigorous treatment of the stability and regularity properties of selective state-space models. Its main theoretical contribution—the formalization of irreversible forgetting—introduces a novel and elegant concept: once a state direction becomes energy-less, it remains so structurally, constraining how future gating can affect the system without violating passivity. This insight not only deepens the theoretical understanding of modern state-space models but also bridges control theory and machine learning in a principled way.

**Weaknesses:**

The paper’s main limitation lies in its narrow empirical validation. While the theoretical framework is rigorous and comprehensive, the experiments are minimal and primarily illustrative rather than demonstrative of broader applicability. As a result, it remains unclear how the proposed theoretical results can be used or extended by the wider machine learning community. The work would benefit from a clearer discussion of its practical relevance, for instance by connecting the theoretical insights to optimization or algorithmic improvements that directly impact model training or design. Without such developments, the contribution risks being perceived as primarily conceptual, with limited immediate value for practitioners working on real-world selective state-space models or related architectures.

**Questions:**

In general, models with long-term memory tend to be more prone to instability. It would be beneficial if the authors could adopt an approach that explicitly analyzes or discusses the stability of such long-memory dynamics.

---

### Official Review · Reviewer_K6Mh · 2025-10-24

**Soundness:** 2
**Presentation:** 2
**Contribution:** 1
**Rating:** 2
**Confidence:** 5

**Summary:**

The work present a rigorous control-theoretic framework based on passivity and Input-to-State Stability (ISS) to analyze selective SSMs. The authors show that even with discontinuous gating, selective SSMs possess well-defined energy structures that ensure exponential forgetting and robust stability. The framework also imposes structural constraints on how gating mechanisms can modify system dynamics.

**Strengths:**

1. Stability analysis for selective SSMs is an interesting topic.

**Weaknesses:**

1. Most theoretic results presented in this paper are well-established in the control literature. For example, the first contribution Thm. 3.1 is the classic Lyapunov theorem for exponential stability, which can be found in most standard nonlinear control textbooks. Furthermore, passivity theory for nonlinear, linear parameter-varying (LPV), and switched (or jump) systems have been extensively studied over the past three decades. Many similar theoretic results can be found in those major control journals and conferences by searching the keywords such as "passivity", "LPV", and "linear switched systems", etc.

2. Passivity is a restricted stability analysis framework due to the specific choice of supply rates. It is useful when the underlying dynamics arise from physical systems with particular choices of input-output pair. For example, mechanical systems are naturely passive when the input is force and the output is velocity. However, even such systems are not passive if the output is position. Since SSMs are abstract dynamical system learned from data, passivity theory will lead to conservative results.  For instance, if an SSM is passive (i.e., satisfying the parametric LMI in Eq. (11)), then there is a strict structural constraint $B(t,x)Q(t)=C(t,x)$. In other words, the off-diagonal terms of (11) has to be zero; otherwise, $L(t,x)\preceq 0$ cannot be guaranteed from Schur's lemma. In practice, such a constraint is rarely enforced or even considered in SSMs.

3. In most control and machine learning applications, the hidden states and input–output variables reside in some real space $\mathbb{R}^n$. Therefore, the generalization to complex space $\mathbb{C}^n$ appears unnecessary and lacks clear motivation.

4. The experimental section is relatively weak and does not provide sufficient empirical support for the theoretical claims. Is there any pre-trained SSM satisfying the passivity condition? Or is there any benefit (e.g., generalization, robustness, training stability) by enforcing passivity on SSMs?

**Questions:**

N/A

---

### Official Review · Reviewer_GMX8 · 2025-10-31

**Soundness:** 3
**Presentation:** 3
**Contribution:** 3
**Rating:** 4
**Confidence:** 4

**Summary:**

This paper introduces a control-theoretic framework for analyzing the stability and regularity of modern Selective State-Space Models (SSMs). Using the ideas of passivity and Input-to-State Stability (ISS), the authors develop a rigorous mathematical foundation that characterizes how energy dissipation and gating mechanisms influence the stability of these input-dependent dynamical systems. Theoretical results include proofs of exponential forgetting, a formulation of irreversible forgetting under discontinuous gating, and a derivation of an LMI-based constraint that ensures passivity. To validate their analysis, the authors conduct small-scale simulations demonstrating that HiPPO-style initialization result in slower energy decay which is consistent with their theory, irreversible forgetting emerges as predicted, and incorporating the proposed LMI regularizer improves training stability in a toy SSM compared to an unregularized baseline.

**Strengths:**

This paper provides a rigorous and well-structured mathematical foundation for analyzing the stability of selective state-space models. The authors present their framework with strong theoretical depth, demonstrating substantial technical effort, and the writing is generally clear and self-contained. The approach is novel, offering a formal and principled way to study the stability and regularity of modern SSMs that rely on input-dependent gating.

**Weaknesses:**

While the paper provides strong theoretical insight, its practical validation is limited. The analysis is supported only by small-scale simulations rather than experiments on real or large-scale SSM architectures. Evaluating the proposed framework on the models that inspired this work, such as Mamba, HGRN, or GLA, would have provided stronger empirical evidence of its relevance and utility. Similarly, a comparison of model performance before and after applying the proposed LMI regularizer would have clarified its practical impact. The focus on continuous-time formulations also limits direct applicability to the discrete-time implementations commonly used in practice. As a result, despite its solid theoretical foundation, it is difficult to assess the real-world effectiveness of the proposed framework.

**Questions:**

1.	The framework is presented for continuous-time selective SSMs. How would the proposed stability and passivity conditions translate to discrete-time implementations such as Mamba or HGRN?
2.	Have the authors considered applying their LMI-based regularizer to existing selective SSMs to verify its effect on stability and performance? If so, would any modifications be required to adapt the regularizer to these architectures?

---

### Note · Authors · 2025-11-13

I have read and agree with the venue's withdrawal policy on behalf of myself and my co-authors.